# Tracing multiple scattering trajectories for deep optical imaging in scattering media

Sungsam Kang [1,2], Yongwoo Kwon[1,2], Hojun Lee[1,2], Seho Kim[1,2], Jin Hee Hong [1,2], Seokchan Yoon [1,2,3] ✉ & Wonshik Choi [1,2] ✉

Multiple light scattering hampers imaging objects in complex scattering media. Approaches used in real practices mainly aim to filter out multiple scattering obscuring the ballistic waves that travel straight through the scattering medium. Here, we propose a method that makes the deterministic use of multiple scattering for microscopic imaging of an object embedded deep within scattering media. The proposed method finds a stack of multiple complex phase plates that generate similar light trajectories as the original scattering medium. By implementing the inverse scattering using the identified phase plates, our method rectifies multiple scattering and amplifies ballistic waves by almost 600 times. This leads to a significant increase in imaging depth—more than three times the scattering mean free path—as well as the correction of image distortions. Our study marks an important milestone in solving the long-standing high-order inverse scattering problems.

In ordinary daily life, we visually perceive the world around us from the light scattered by objects. Our brain processes the received signals assuming that they are scattered only once from the surface of the objects[1]. In situations where the objects are embedded deep within a scattering medium, almost all the received signals are scattered multiple times. Without knowing their travel history, our brain cannot process them properly, thereby perceiving the objects as obscured. Examples include automobiles moving on a foggy day, abnormal cells hidden under the skin tissues, and nervous systems under the cranial bone[2]. For this reason, imaging modalities actively used in real practices aim to suppress multiple scattering for finding ballistic waves traveling straight through the scattering medium. However, the exponential attenuation of ballistic waves with depth sets a hard limit on their achievable imaging depth[3,4]. To go beyond this limit, it is necessary to make use of multiple scattering for image reconstruction. This requires finding the trajectories of multiply scattered waves, which will allow for implementing the inverse scattering to see the objects clearly as if there is no scattering medium in the first place (Fig. 1). However, it is extremely difficult to trace individual light trajectories, especially when the objects are completely embedded in a thick and bulk scattering medium. In this general situation, we should select and trace only those backscattered waves that make roundtrips

to the embedded target objects while ruling out the others that travel to shallower depths.

There have been many prior reports making the use of multiple scattering for image reconstruction. However, almost all the previous works handled the problem in limited conditions. In most cases, the target object is in free space on the opposite side of either a scattering layer[5–7] or a wall[8,9] rather than embedded within a scattering medium. Some studies demonstrated a wave focusing on a target in a scattering medium[10–12]. However, wave focusing leads to imaging only for a thin scattering layer, where a focus can be scanned within the so-called memory effect range[10,13]. The thicker the scattering medium, the narrower the memory effect range, which is no longer usable for imaging.

Tracing multiple scattering trajectories in general situations of imaging an embedded target is the task of solving an inverse scattering problem[14]. In this context, imaging modalities can be classified by the number of scattering events that they can trace, which corresponds to the order of Born approximation they attempt to address[15]. This classification essentially involves determining the number of unidentified layers the modalities seek to map. The first-order approach finds the light scattered only once by the object of interest[16,17]. Almost all microscopic imaging modalities relying on ballistic waves fall into this category. They employ various gating operations[18–24] to filter out the

[1]Center for Molecular Spectroscopy and Dynamics, Institute for Basic Science, Seoul, Korea. [2]Department of Physics, Korea University, Seoul, Korea. [3]School of Biomedical Convergence Engineering, Pusan National University, Yangsan, Korea. ✉e-mail: sc.yoon@pusan.ac.kr; wonshik@korea.ac.kr

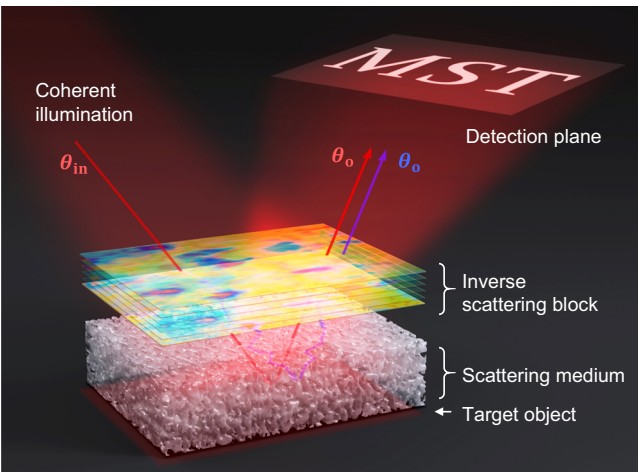

**Fig. 1 | Illustration of the inverse scattering.** A scattering medium can be made transparent by placing a virtual inverse-scattering block counteracting the scattering medium. A target object, which is the letters "MST," located under the scattering medium becomes visible because of the inverse-scattering block. The red and blue arrows indicate the ballistic and multiply scattered waves, respectively.

multiply scattered waves. High-order approaches have been developed to trace multiple scattering for extending imaging depth; however, they can trace only a limited number of scattering events. For instance, adaptive optics, which deals with the perturbation of either the excitation or returning beams, can be viewed as second- or third-order methods[25–34]. This classification is due to their attempt to identify the input/output pupil layers in addition to a sample layer. Approaches based on the memory effect can be classified as a similar category[13,35,36]. In imaging an object embedded within a scattering medium, tracing beyond the third order has not been possible thus far mainly because of insufficient sampling and a lack of a physics model to address the underdetermined nature of the problem, especially in the presence of strong multiple scattering with no interaction with the objects of interest.

Here, we propose a method—multiple scattering tracing (MST) algorithm—that can keep track of a number of scattering events responsible for reconstructing an object embedded deep within a thick scattering medium, using the experimentally measured reflection matrix of the scattering medium[30,31]. The algorithm uses the intrinsic correlations of multiple scatterings to find a set of complex phase plates that reproduce their trajectories in the scattering medium enclosing the target objects. Through the inverse of the transmission matrix of the phase plates, which is conceptually equivalent to placing an inverse scattering block counteracting the scattering medium, we can rectify the multiple scattering and obtain a diffraction-limited image of the object as if there is no scattering medium (as illustrated in Fig. 1). Essentially, this process realizes the deterministic use of multiple scattering for microscopic image formation by converting the multiply scattered waves to ballistic signal waves. We validated the proposed concept using numerical simulation and experimentally demonstrated its performance by conducting in vivo imaging of a mouse brain under an intact skull, an extreme form of a scattering medium. With the MST algorithm, we could identify up to, but not limited to, eight layers of phase plates representing thick skull tissue and a layer of underlying target biological structures. Considering that each phase plate and the target structure can induce scattering events, this corresponds to tracing a total of 17 scattering events—eight on the way in, one induced by the target structure, and another eight on the way out. In these demonstrations, the ballistic signal waves were amplified almost 600 times by converting the multiply scattered waves that are otherwise considered noise. This effectively reduces the

optical thickness of the scattering medium by more than three times the scattering mean free path. As a result, the imaging depth increases by the same factor of the scattering mean free paths. Our work marks an important breakthrough in solving the high-order inverse scattering problem in the general situation when strong multiply scattered waves exist without any interaction with the target object.

## Results

### Working principle

Let us consider probing an object embedded within a scattering medium using a light wave with a specific incidence angle $\theta_{in}$ (Fig. 2a). While there is a tiny fraction of the incident wave, termed as a ballistic wave, that preserves its propagation direction in the scattering medium (red arrows), the majority are scattered multiple times on their way to and from the object of interest (blue arrows). This multiple scattering leads to random spreading of the propagation angles, thereby undermining the reconstruction of the image information. In the context of imaging, one can find the object spectrum from the momentum difference $\Delta k(\theta_o; \theta_{in}) = k_0 \sin\theta_o - k_0 \sin\theta_{in}$ for a ballistic wave, where $k_0 = 2\pi/\lambda$, $\lambda$ is the wavelength of the light source, and $\theta_o$ is the angle of the backscattered wave[22]. However, $\Delta k(\theta_o; \theta_{in})$ of the multiply scattered wave differs from the object spectrum because the actual incident and reflected angles of the object are $\theta_{in}^M \neq \theta_{in}$ and $\theta_o^M \neq \theta_o$, respectively. The multiple scattering process is random but deterministic. If one can keep track of the multiple scattering trajectories and, thus, find $\theta_{in}^M$ and $\theta_o^M$, then the object spectrum can be obtained from $\Delta k(\theta_o^M; \theta_{in}^M)$. However, this is a heavily under-determined problem because numerous possible trajectories generate the same $\theta_o$. Therefore, in most studies, only the ballistic waves therein are exploited for image reconstruction[25–28,30].

To keep track of the multiple scattering trajectories and use the multiply scattered waves for image reconstruction, we set up two strategies: experimental recording of a time-gated reflection matrix, and the development of the MST algorithm that finds $\theta_{in}^M$ and $\theta_o^M$ from the measured reflection matrix. In the first strategy, we measure the electric fields of the backscattered waves, including both the ballistic wave and multiply scattered waves, whose flight times are distributed within a finite time-gating window of 100 fs, given by the pulse width of the laser. This eliminates a large fraction of scattering events that have no interaction with the target objects located at the depth of interest. These include backscattering at shallow depths. By repeating the same measurements for all the possible incident angles, we can construct a reflection matrix $\boldsymbol{R}$ that describes the interaction between a light wave and scattering medium (see Methods and Supplementary Section Experimental setup for laser scanning reflection-matrix microscopy for a detailed experimental setup and the construction of $\boldsymbol{R}$.).

As a second strategy, we propose a powerful MST algorithm that exploits the wave correlation in the measured $\boldsymbol{R}$. Using this algorithm, we can model the thick and inhomogeneous volumetric scattering medium as a discrete stack of thin transmissive phase plates that generates light trajectories similar to those generated by the original scattering medium (Fig. 2b). In this case, we assume that the forward scattering is dominant and ignore the reflections and absorptions in the scattering medium. In fact, time-gated detection plays a critical role in guaranteeing the validity of this assumption, because waves experiencing multiple reflections back and forth inside the scattering medium tend to have much more elongated flight times than the ballistic waves and forward scattering components that interact with the target object. In this multilayer model, a light wave is assumed to undergo a spatially varying phase shift $\varphi_k(\boldsymbol{\rho})$ at each $k^{th}$ layer located at a depth $z = z_k$ with $\boldsymbol{\rho} = (x, y)$, which is the lateral coordinate, and then propagate through free space to the adjacent layer. Therefore, the inward propagation of the wave from the surface of the scattering medium at $z_N$ to the object plane at $z_O$ is described by a transmission matrix $\boldsymbol{T} = \prod_{k=1}^N \boldsymbol{T}_k = \boldsymbol{T}_1 \boldsymbol{T}_2 \boldsymbol{T}_N$, where $\boldsymbol{T}_k = \boldsymbol{P}_{z_{k-1}, z_k} \boldsymbol{\Phi}_k$ is the transmission

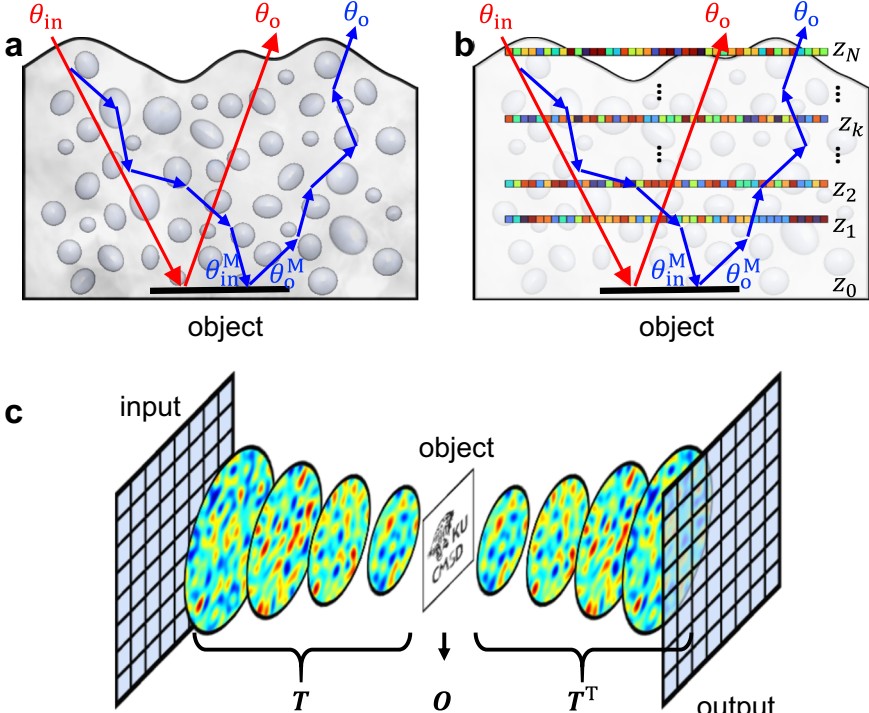

**Fig. 2 | Working principle of the MST algorithm. a** Roundtrip travel of waves to an object embedded within a scattering medium. Multiply scattered waves (blue arrows) change propagation directions multiple times, whereas the ballistic wave (red arrows) preserves its propagation direction when propagating through the scattering medium. **b** Modeling of the scattering medium by a discrete stack of phase plates. Each phase plate is described by a phase function $\varphi_k(\boldsymbol{\rho}_j)$ at a depth $z_k$. The object of interest is located at $z_0$. **c** Illustration of our forward model describing the reflection matrix $\boldsymbol{R}$ with a stack of discrete phase plates. For better understanding, the reflection process by the target object is shown to the right of the target.

matrix of the $k^{\text{th}}$ layer; $\boldsymbol{P}_{z_{k-1},z_k}$ is the free-space propagation matrix for the waves traveling from a plane $z_k$ to a plane $z_{k-1}$; and $\boldsymbol{\Phi}_k$ is the diagonal matrix, whose elements are given by $\exp[i\varphi_k(\boldsymbol{\rho})]$ (Fig. 2c). Considering the roundtrip travel of the light wave, the reflection matrix of a scattering medium is modeled in the space domain as

$$\boldsymbol{R}^{(\text{M})}_{z_N,z_N} = \left(\boldsymbol{T}^{\text{T}}_N \cdots \boldsymbol{T}^{\text{T}}_2 \boldsymbol{T}^{\text{T}}_1\right) \boldsymbol{O}\left(\boldsymbol{T}_1 \boldsymbol{T}_2 \cdots \boldsymbol{T}_N\right) = \boldsymbol{T}^{\text{T}} \boldsymbol{O} \boldsymbol{T}, \quad (1)$$

where $\boldsymbol{O}$ is a diagonal matrix describing the amplitude reflection $O(\boldsymbol{\rho},z_0)$ of the object of interest located at the object plane, and $\boldsymbol{T}^{\text{T}}$ is the transpose of $\boldsymbol{T}$, which accounts for the outward propagation of the reflected waves from the object back to the surface of the scattering medium. Essentially, there are $2N$ phase layers and an object layer in $\boldsymbol{R}^{(\text{M})}_{z_N,z_N}$ that describe the roundtrip of the waves entering the surface of the scattering medium at $z_N$, all the way to the object, and returning back to the surface. Our task is to find a set of $\varphi_k(\boldsymbol{\rho})$ and $O(\boldsymbol{\rho},z_0)$ from the experimentally measured $\boldsymbol{R}_{z_N,z_N}$. Finding $\varphi_k(\boldsymbol{\rho})$ is identical to identifying $\boldsymbol{T}$, which in turn provides information on $\theta^{\text{M}}_{\text{in}}$ and $\theta^{\text{M}}_{\text{o}}$. Multiplying the inverses of $\boldsymbol{T}$ and $\boldsymbol{T}^{\text{T}}$ on the input and output sides of the measured $\boldsymbol{R}_{z_N,z_N}$, respectively, we can compensate for the multiple scattering effects, which is equivalent to placing the virtual inverse-scattering block as shown in Fig. 1.

Our MST algorithm consists of two major steps. The first step is to access each $k^{\text{th}}$ layer in $\boldsymbol{R}_{z_N,z_N}$, and the second step is to find $\varphi_k(\boldsymbol{\rho})$ asymptotically by taking advantage of the correlations of the wave fields interacting with the object. By repeating this process for all the $2N$ layers iteratively, we gradually reach the ground-truth $\varphi_k(\boldsymbol{\rho})$. We describe the detailed process of the MST algorithm in the Method section and Supplementary section Inverse scattering model. As a result of the MST algorithm, we obtain the reconstructed phase map $\varphi^{\text{c}}_k(\boldsymbol{\rho}_j)$ of each layer. Then the correction transmission matrix $\boldsymbol{T}^{\text{c}}$ is constructed by $\varphi^{\text{c}}_k(\boldsymbol{\rho}_j)$, which yields the light trajectory. By multiplying the inverses of $\boldsymbol{T}^{\text{c}}$ and $(\boldsymbol{T}^{\text{c}})^{\text{T}}$ on the input and output sides of the measured $\boldsymbol{R}$, respectively, we obtain a corrected reflection matrix, $\boldsymbol{R}^{\text{c}} = [(\boldsymbol{T}^{\text{c}})^{\text{T}}]^{-1}\boldsymbol{R}(\boldsymbol{T}^{\text{c}})^{-1}$ that compensates for the multiple scattering effects. Finally, the object function $O(\boldsymbol{\rho}_i,z_0)$ is reconstructed from the diagonal elements of $\boldsymbol{R}^{\text{c}}_{z_0,z_0}$ (see Supplementary Section Inverse scattering model for the flow chart describing the detailed iteration procedures).

We would like to emphasize that our MST algorithm is specially designed for deep imaging in a thick scattering medium. The incident wave experiences substantial lateral spreads throughout the thick and bulk scattering medium, whereas the backscattered waves are recorded only for the limited field of view. Therefore, there are substantial multiple scattering components in $\boldsymbol{R}$ that have no interaction with the object of interest, and thus, cannot be modeled by $\boldsymbol{R}^{(\text{M})}$. This renders the direct minimization of the difference between $\boldsymbol{R}$ and $\boldsymbol{R}^{(\text{M})}$, a strategy mainly employed in the conventional approaches for addressing higher-order inverse scattering problems in isolated single cells[37,38], highly under-determined and ill-posed. Instead, our algorithm aims to find those wave components in $\boldsymbol{R}$ that are responsible for the object image reconstruction. Based on wave correlation, the algorithm identifies the phase functions at multiple depths to coherently accumulate the object-information-carrying multiple scattering signal components. This novel approach is highly robust to the existence of multiple scattering that remains unaccounted for by the model that does not carry object information. Furthermore, its computational cost can be much lower than the minimization algorithm. As shown below, it takes only 10 min to trace nine scattering events generated by four phase plates and one object plane when the sampling area is $100 \times 100$ pixels (field of view (FOV) size: ~$50\lambda \times 50\lambda$).

## Proof of concept with numerical simulation

We performed numerical simulations to validate our proposed method of solving the inverse scattering problem (Fig. 3). Here, we consider a case where a target object is covered with multiple discrete phase plates whose phase functions are known a priori (Fig. 3a). Four phase plates are placed at depths of $\{z_{k=1..4}\} = \{35,50,70,100\}$ μm, respectively, from the target object shown in the inset in Fig. 3a. The phase functions $\varphi_k$ of the four layers are shown in Fig. 3b. We set the size of each phase map as $200 \times 200$ μm², while that of the target object is set as $45 \times 45$ μm². Each phase map is filled with a random phase pattern to mimic a realistic scattering medium. In addition, different polygonal patterns and numbers are superimposed on each layer for ease of performance evaluation of the proposed algorithm (see Supplementary Section Numerical demonstration of MST algorithm with randomly patterned layers for the case of phase plates filled only with random phase patterns). Then, we numerically generate the reflection matrix $R$ of the sample at a wavelength of 900 nm and an angular coverage of 1.0 NA based on Eq. (1).

By applying the MST algorithm to $R$, we reconstructed the phase functions $\varphi_k^c$ for the four scattering layers, as shown in Fig. 3c (see Supplementary Section Numerical study for the detailed iteration process of the MST algorithm). Essentially, our algorithm located nine scattering events, considering the roundtrip and the scattering at the object. Note that a circular sub-area of each phase function was recovered owing to the intrinsic cone-shaped imaging geometry (Methods and Supplementary Section Sampling area of the phase functions). The numbers and polygonal patterns in each phase function were clearly identified, proving the efficacy of the proposed algorithm. By taking the diagonal elements of $R_{z_0,z_0}$, we could obtain a

conventional confocal reflectance image of the object as shown in Fig. 3d[30]. Due to the multiple scattering by the upper scattering medium, the detailed object structure was invisible. We constructed $T^c$ using the identified phase functions $\varphi_k^c$ and obtained the corrected reflection matrix $R^c$ by applying the inverses of $T^c$ and $(T^c)^T$ to the input and output sides of $R$, respectively. Then, the scattering-free object image was reconstructed from the diagonal elements of $R_{z_0,z_0}^c$, which is referred to as an MST image. Figure 3e displays the reconstructed MST image, showing an excellent agreement with the ground-truth object image depicted in Fig. 3a. To quantify the accuracy of the proposed algorithm, we calculated the Pearson correlation coefficients of each phase function $\varphi_k^c$, the transmission matrix $T^c$, and the MST image with their ground-truth counterparts. The average correlation was measured to be approximately 70–80% for each phase function and transmission matrix while that of the object image was about 94%. The discrepancy mainly arises because not all the scatterings can be accounted for. In fact, the spatial resolution of mapping the phase functions is reduced with an increase in the distance from the object plane (Supplementary Section Spatial resolution of identifying phase function $\varphi_k$). The correlation of the object image was much higher than that of the others since the confocal gating suppressed the unaccounted multiple scattering.

The rectification of multiple scattering by the MST algorithm makes the scattering medium transparent. To quantify this effect, we displayed an angular spread function (ASF), which is the angular distribution of a normally incident plane wave with depth. Figure 3h shows the ASF measured after each layer before the multiple scattering rectification. The ASF was progressively broadened as it propagated through the scattering layers, and the ballistic wave (peak in the

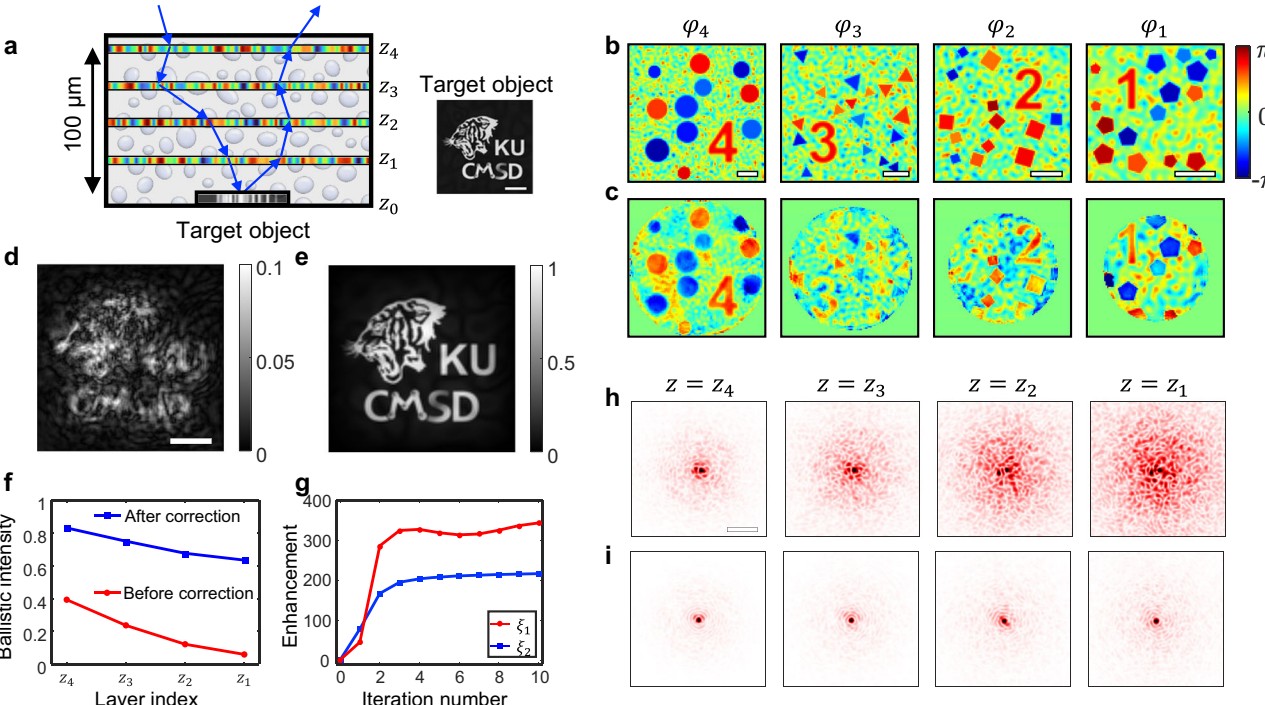

**Fig. 3 | Numerical demonstration of tracking multiple scattering trajectories.**
**a** Schematic of the sample configuration. Four phase plates are placed on top of a target object at a distance of $\{z_{k=1..4}\} = \{35,50,70,100\}$ μm, respectively, from the target object at $z_0 = 0$. Inset: ground-truth target object. Scale bar: 10 μm. **b** Ground-truth phase functions $\varphi_k(\rho)$ of the four scattering layers in **a**. The side lengths of the phase maps are (100, 130, 160, 200) μm from the left. Scale bars: 30 μm. **c** Identified phase functions $\varphi_k^c(\rho)$ by MST algorithm. **d** Conventional confocal reflectance image of the target. Scale bar: 10 μm. **e** MST image by rectifying the identified multiple scattering trajectory. Color scales in **d**, **e** are normalized by the

maximum amplitude in **e**. **f** Ballistic wave intensity of the normally incident plane wave measured after each phase plate before (red circular dots) and after (blue square dots) the application of MST algorithm. **g** The performance of MST algorithm by evaluating $\xi_1$ and $\xi_2$ with the iteration process. **h** Angular spread function of a normally incident plane wave in the spatial frequency domain measured underneath each phase plate. **i** Same as **h**, but after rectifying the multiple scattering trajectory. Angular spread functions are displayed in the spatial frequency coordinates $(k_x, k_y)$ with its center corresponding to (0,0). Scale bar: $0.1 k_0$.

center) preserving the original incident angle was drastically attenuated (red circular dots in Fig. 3f). When we rectified the multiple scattering trajectories by canceling $\varphi_k$, the broadening of ASF was largely removed (Fig. 3i), implying that a large fraction of the multiple scattering was converted to a ballistic signal wave. The ballistic wave intensity was attenuated much less after the multiple scattering rectification (blue square dots in Fig. 3g), with an increase of 18.5 times in its final intensity. Considering the roundtrip, there was $\xi_1 = 344$ times enhancement of the ballistic signal. In terms of the scattering mean free path $l_s$, the four phase plates can be considered a scattering medium with an optical thickness of $3.3l_s$, but their thickness changed to $0.42l_s$ after the rectification. Here, scattering mean free path was estimated by the intensity ratio between the diagonal elements and the total intensity in the spatial frequency domain transmission matrix.

The enhancement $\xi_1$ of the ballistic signal wave can be explained by the $T^c$ identified by the MST algorithm. One can estimate the attenuation of the ballistic wave on its way to the object, as: $\eta_T = \sum_i |\widetilde{T}^c_{ii}|^2 / \sum \sum_{i,j} |\widetilde{T}^c_{ij}|^2$; here, $\widetilde{T}^c_{ij}$ is the $(i,j)^{th}$ matrix element of $\widetilde{T}^c$ in the spatial frequency domain obtained by the Fourier transform of $T^c$. Because the multiple scattering rectification is the action of multiplying the inverse of $T^c$, there is an increase in the ballistic signal by a factor of $1/\eta_T$ during the one-way propagation. Therefore, the roundtrip enhancement of the ballistic signal is given by $\xi_1 = \eta_T^{-2}$. The behavior of $\xi_1$ is shown in Fig. 3g with the iteration process of MST algorithm, indicating its steady increase and saturation to $\xi_1 = 344$. This analysis confirms that our proposed method made the deterministic use of multiple scattering for the image reconstruction; this is a clear distinction from the approaches based on Born approximation and conventional adaptive optics.

Multiple scattering rectifications increase the ballistic signals and attenuate the multiple scattering background signals. This jointly enhances the signal-to-background ratio of the confocal image after the application of MST algorithm, and thus, the achievable imaging depth. Here, we introduce another criterion $\eta_c$ that quantifies the ballistic signal enhancement in the reconstructed MST image: $\eta_c = \sum_i |R^c_{ii}|^2 / (\sum_i |R_{ii}|^2)$. Here, $R_{ii}$ and $R^c_{ii}$ are the diagonal elements of $R_{z_0,z_0}$ and $R^c_{z_0,z_0}$, respectively. Essentially, the $\eta_c$ indicates the

enhancement of the confocal signal, which is primarily related to the enhancement of the ballistic signal. However, the diagonal elements of $R_{z_0,z_0}$ contain substantial multiple scattering components when the initial ballistic signal is excessively weak. Therefore, $\eta_c$ underestimates the actual enhancement of the ballistic signal. For a better estimation of the ballistic signal enhancement, we introduce another parameter $\alpha$, representing the ballistic signal's contribution in the diagonal element of $R$. Then, $\xi_2 = \eta_c/\alpha$ is the ballistic signal enhancement obtained from the MST image analysis (see Supplementary Section Quantitative analysis of the performance of the MST algorithm for details). The behavior of $\xi_2$ is shown in Fig. 3g with iteration; first, it increased and then saturated to a finite value of 217, which means that the ballistic signal in the MST image shown in Fig. 3e was increased by $\xi_2 = 217$ times with respect to the initial confocal image shown in Fig. 3d. $\xi_2$ provides the direct measure of the enhancement of image quality, but there is an ambiguity in estimating $\alpha$, especially when the initial ballistic signal is extremely weak. In this case, $\xi_1$ is a more reliable measure of ballistic signal enhancement than $\xi_2$.

## Experimental validation of the MST algorithm

We experimentally validated the MST algorithm using multi-layered onion tissue as a scattering medium. As illustrated in Fig. 4a, we placed an 800-μm-thick onion tissue on a custom-made resolution target. The onion tissue contained layers of cellular structures, which allowed us to check whether the phase functions retrieved by the MST algorithm originated from the real structures. The resolution target was fabricated by depositing a 200-nm-thick layer of gold onto a glass substrate with a patterned photomask. The target pattern was composed of multi-scale Siemens star structures. Using laser scanning reflection-matrix microscopy[30,31], we recorded the time-gated reflection matrix $R_{z_0,z_0}$ of the sample with the focal plane of the objective lens set to the axial position of the target ($z_0$) (see Supplementary Section Experimental setup for laser scanning reflection-matrix microscopy for experimental setup and matrix construction). As shown in Fig. 4b, the time-gated confocal image reconstructed by the diagonal elements of $R_{z_0,z_0}$ was highly distorted due to the multiple scattering and aberrations induced by the onion tissue. Fine microscopic structures were completely obscured

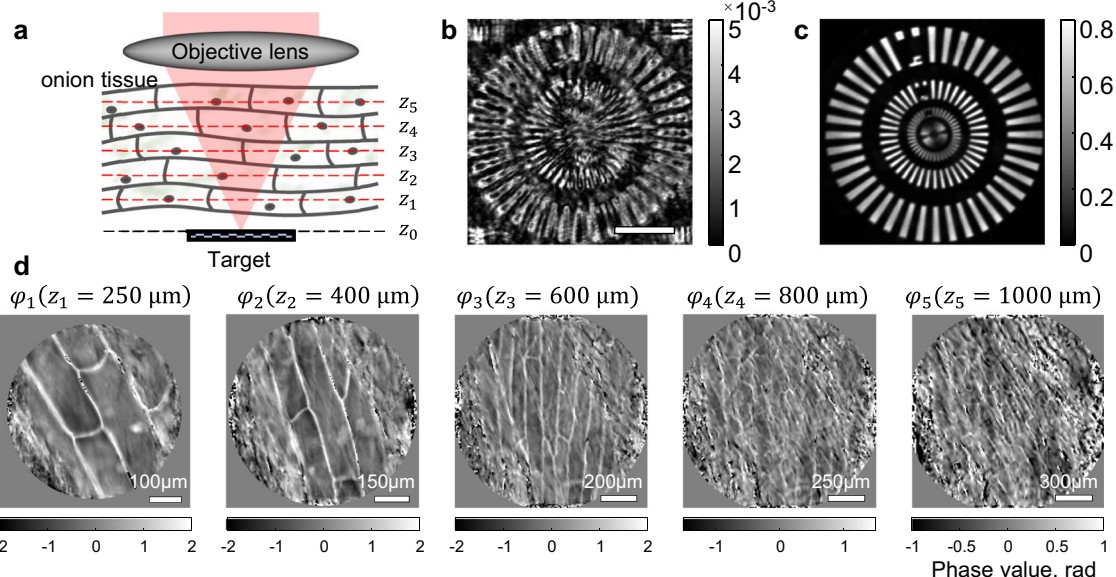

**Fig. 4 | Experimental demonstration of MST algorithm for a volumetric scattering medium. a** Schematic of the sample configuration. An 800-μm-thick onion tissue was placed on top of a custom-made resolution target, and a time-gated reflection matrix was recorded with its focus and temporal gating set to the target depth. **b** Intensity map of confocal reflectance image. Scale bar, 30 μm. **c** MST image after rectifying the identified multiple scattering trajectories. Intensity maps in **b**, **c** are normalized by the peak intensity of **c**. **d** Identified phase functions, $\varphi^c_k(\boldsymbol{\rho})$, at five different depths by the MST algorithm. Color bars, phase in radians.

while large features remained visible to some extent due to the residual ballistic signals (see Supplementary Section Quantitative analysis of the performance of the MST algorithm for detailed PSF analysis). Then, we applied MST algorithm by modeling the thick onion tissue as five discrete layers placed at $\{z_{k=1..5}\} = \{250, 400, 600, 800, 1000\}$ μm from the target plane at $z_0 = 0$. By rectifying the identified multiple scattering trajectories, we obtained the object function and reconstructed the MST image (Fig. 4c). We could identify the fine details of the target object with ~800 nm spatial resolution, close to the diffraction-limited resolution of 650 nm for the illumination wavelength of 1300 nm. Phase functions $\varphi_k^c(\boldsymbol{\rho})$ of the five layers obtained by the MST algorithm are displayed in Fig. 4d. The area of the reconstructed phase functions increased with an increase in the distance from the target plane due to the cone-shaped imaging geometry. The walls of individual onion cells and their nuclei were clearly visible, especially in $\varphi_1^c$, $\varphi_2^c$, and $\varphi_3^c$. This clearly demonstrates that the MST algorithm reconstructs valid trajectories of multiple scattering by the real structures of the scattering medium covering the target object. Cellular shapes in the layers at $z_4$ and $z_5$ were rather indistinctive because the spatial resolving power for retrieving these layers was not high enough to identify the cell walls (see Supplementary Section Spatial resolution of identifying phase function φk for a detailed discussion of the lateral and axial resolutions of the phase function reconstruction). The enhancement of ballistic signal estimated by $\varphi_k^c$ was $\xi_1 = 343$.

## Demonstration of MST algorithm with a highly scattering skull tissue

So far, we have validated the MST algorithm numerically and experimentally with a layered scattering medium. Next, we demonstrate our method with a thick bulk scattering medium which does not have well-defined layered structures. For the demonstration, we measured the reflection matrix $\boldsymbol{R}_{z_0, z_0}$ of the resolution target under a 180 μm-thick cranial bone of a mouse, as illustrated in Fig. 5a. As shown in Fig. 5b, the intensity map of the conventional confocal reflectance image was distorted due to the scattering and the aberration by the thick skull tissue. Unlike the previous samples, the axial positions and the number of required phase plates were not well defined for this bulk cranial tissue. We first determined the approximate position of the skull tissue by measuring the ballistic enhancement $\xi_2$ of the MST algorithm assuming a single phase plate. We scanned the axial position of the phase plate from 50 μm to 400 μm and found that the skull tissue was positioned approximately 100 μm ~ 300 μm away from the resolution target (blue shaded region in Fig. 5c). Then, we applied the MST algorithm by increasing the number of the phase plates $N$ whose axial positions are shown in Fig. 5d.

The resulting MST images and the map of the phase plates are displayed in Fig. 5e. For a single phase plate (top row of the figure), only the lower half of the resolution target was visible, and the overall intensity was much lower than the MST images with many phase plates. This is because only a small fraction of the multiple scattering

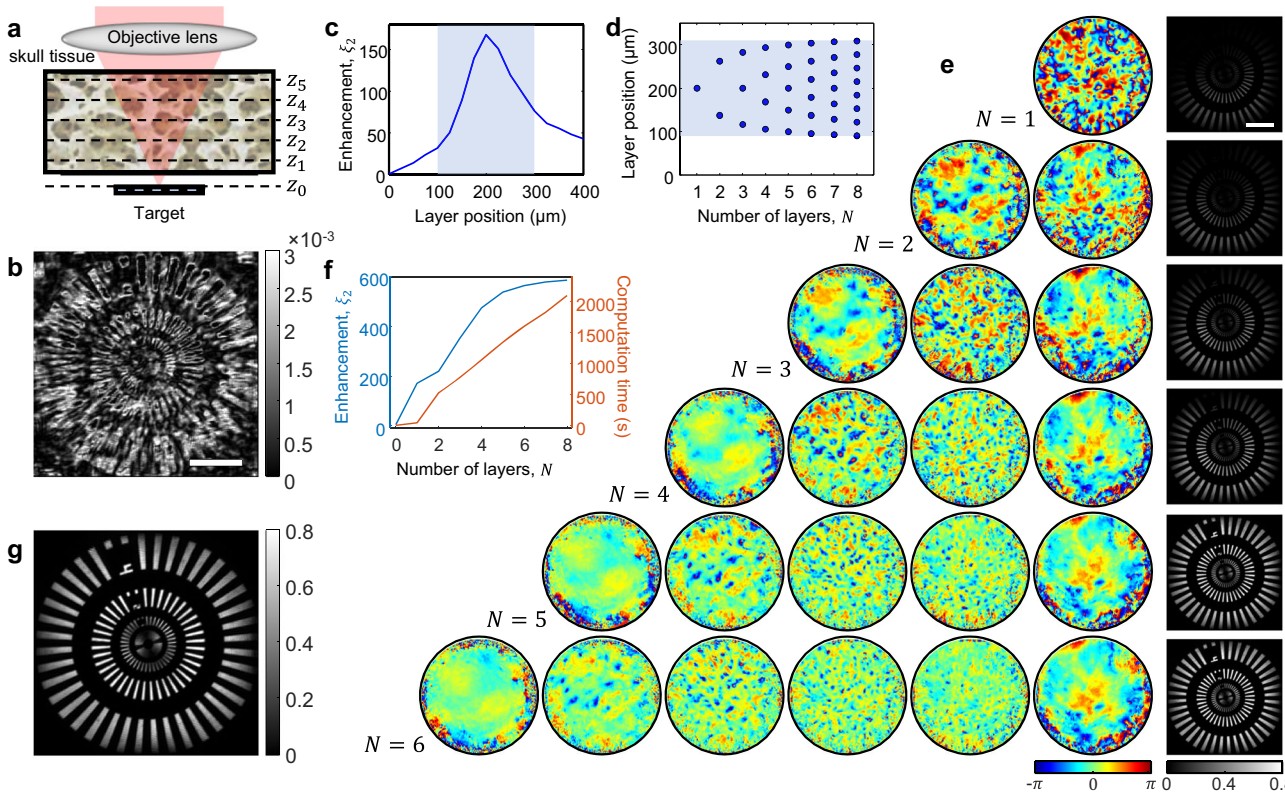

**Fig. 5 | MST images depending on the number of phase plates for a bulk scattering tissue. a** Schematic of the sample configuration. A 180-μm-thick excised mouse skull was placed on top of a custom-made resolution target. **b** Intensity map of the conventional confocal reflectance image of the resolution target. Scale bar, 20 μm. Color bar, normalized by the peak intensity of **g**. **c** Ballistic enhancement $\xi_2$ by MST for a single phase plate ($N = 1$) forward model depending on the axial position of the plate. The shaded region corresponds to the approximate depth ranges of the skull tissue. **d** Positions of the phase plates set in the MST algorithm with the increase of the number of phase plates, $N$. **e** Phase maps, and MST images depending on $N$. MST images were normalized by the peak MST intensity for $N = 6$. Color bar for the phase plates: phase in radians. The diameter of each phase plate is given by $L_k \simeq 1.2z_k + L_0$, where $z_k$ is shown in **d** and $L_0 = 84$ μm. **f** Ballistic enhancement $\xi_2$ and corresponding computation time as a function of $N$. **g** Intensity map of MST image with $N = 5$. Color bar, normalized by the peak intensity.

trajectories could be corrected with a single phase plate. With the increase in the number of phase plates, the MST image became sharper over the entire view field, and the overall intensity was substantially increased. For quantitative analysis, we estimated the ballistic enhancement $\xi_2$ with the increase of $N$ up to 8 layers (Fig. 5f). The ballistic signal rectified by the MST was enhanced by almost 580 times the initial ballistic intensity. The increase in $\xi_2$ was saturated around $N = 5$. Considering that the computation time of the MST algorithm is proportional to $N$, the optimum number of phase plates without compromising the image contrast would be around $N = 5$. Note that the required computation time is for post-processing and, thus, does not affect the experimental recording of the reflection matrix. If necessary, we can speed up the computation time by reducing the region of interest (ROI) (see Supplementary Section Computational cost). In Fig. 5g, we display the final MST image with $N = 5$ phase plates. It shows fine details of the target with high contrast despite the strong multiple scattering. The corresponding phase functions in Fig. 5e show real cellular structures in the cranial tissue as multiple blue spots with negative refractive index contrast. In the following section, we prove that these structures are osteocytes of the skull from in vivo imaging of the mouse brain.

Let us elaborate on the relation between the number of traced scattering events and the scattering mean free path. The scattering mean free path is the average distance between successive scattering events. At each phase plate, there's a certain probability of scattering occurring or not. Therefore, despite a maximum of 17 scattering events, the average number of scattering events could be significantly fewer. For instance, the derived phase plates from mouse cranial tissue shown in Fig. 5 correspond to an average scattering length of $6.4l_s$ in the round-trip, which is much less than 17. The rectification of the multiple light

scattering events, based on these phase plates, reduces the optical thickness by the equivalent number of the scattering mean free paths.

## Application to in vivo imaging

Here, we demonstrate that the proposed MST algorithm works for natural objects with weak reflectance embedded in a heavily scattering medium. We conducted in vivo imaging of a living mouse brain with its skull intact. We then applied our MST algorithm to find the phase functions of the skull and reconstruct myelinated axons in the cortical brain. An adult mouse (five-month-old) was placed on a custom-built sample stage after removing its scalp and covering the center of the exposed parietal bone with a circular glass window (Fig. 6a). The skull thickness was about 200 μm. We measured the time-gated reflection matrix $\boldsymbol{R}_{z_0, z_0}$ at a depth of 270 μm from the surface of the skull over $112 \times 112$ μm² ROI. Then, we applied MST algorithm by modeling the skull as five discrete layers placed at the distances of $\{z_{k=1...5}\} = \{140, 180, 220, 260, 300\}$ μm from object plane at $z_0$ (Fig. 6b). The diameters of the obtained phase functions were 280, 330, 380, 430, and 480 μm, respectively. Figure 6c shows the phase functions $\varphi_1^c$, $\varphi_2^c$, and $\varphi_3^c$ retrieved by MST algorithm. In the individual phase functions, many blue spots whose diameter is 10 – 15 μm are visible. In fact, they correspond to the osteocytes in the skull, presenting the negative phase delays relative to the background skull tissue due to their smaller refractive index. To validate this, we compared the phase functions with the MST images recorded directly at the same depths inside the skull (Fig. 6d). The volume segmentation was applied to the osteocyte cell bodies in the MST images for ease of comparison. As indicated by numbers, the location of osteocytes in the phase functions found by our MST algorithm agrees well with their actual positions directly measured at the corresponding depths.

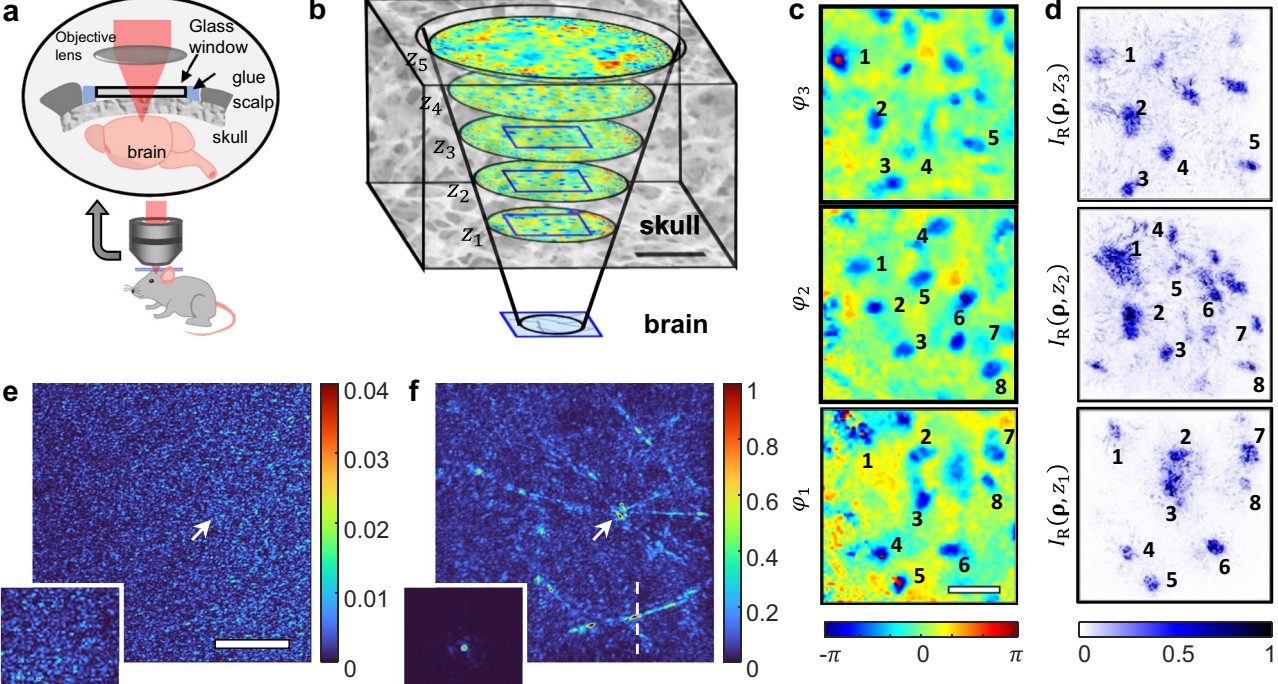

**Fig. 6 | In vivo imaging of a mouse brain of a matured mouse through the intact skull. a** Schematic of the sample configuration. **b** Phase functions, $\varphi_k^c(\boldsymbol{\rho})$, at five different depths identified by the MST algorithm for a reflection matrix measured at a depth of 270 μm below the surface of the skull. $\{z_{k=1...5}\} = \{140, 180, 220, 260, 300\}$μm from the object plane. Scale bar: 100 μm. **c** Identified phase functions $\varphi_{1-3}(\boldsymbol{\rho})$ at the depths $z_{1-3}$ in the blue boxes in **b**. Scale bar: 30 μm. Color bar, phase in radians. **d** MST images obtained by the reflection

matrices measured directly at the depths $z_{1-3}$. Color bar, normalized intensity. The blue dots in **d** correspond to osteocyte cell bodies. Numbers in **c** and **d** in the same depth refer to the same osteocyte cells. **e, f** Confocal reflectance image and MST image of the mouse brain at a depth 270 μm from the surface of the skull, respectively. Scale bar: 30 μm. Insets show the PSFs at points indicated by white arrows. Scale bar: 10 μm. The images were normalized by the peak intensity in **f**.

Figure 6e, f show the resulting confocal and MST images, respectively. Fine myelinated axons in the brain cortex are clearly visible in the MST image, while there were no discernible microscopic structures in the confocal image. Insets in Fig. 6e, f show the representative PSFs for a focused illumination at positions indicated by white arrows. The PSF in Fig. 6e is completely diffused owing to the severe multiple scattering induced by the skull. On the contrary, the PSF became single peaked after applying the MST algorithm (inset in Fig. 6f). The full width at half maximum (FWHM) of the PSF after correction ( ~ 2 μm) was broader than the diffraction-limited resolution of 0.65 μm. However, this does not imply that the spatial resolution of the MST image is 2 μm. The PSF shown in the inset of Fig. 6f is the wide-field image under a focused illumination. The MST image is reconstructed by the diagonal part of $R_{z_0,z_0}^c$. This acts as a confocal filter, which raises the spatial resolution. For instance, the FWHM of the cross-sectional profile of MST image across a myelinated axon (white dashed line in Fig. 6f) was about 1.05 μm, which means that the resolving power is better than this myelin width.

We also investigated the performance of the MST algorithm and the behavior of the PSFs as a function of the number of the phase plates $N$ (see Supplementary Section Behavior of multiple scattering rectification by MST algorithm in in-vivo imaging). This led us to prove that its performance is far better than any previously developed imaging modalities relying on ballistic waves (see Supplementary Section Comparison with existing imaging modalities).

## Discussion

In this study, we proposed a method to trace the multiple scattering trajectories in situ and convert them into ballistic signal waves for imaging objects of interest embedded within a scattering medium as if there were no scattering medium. Conventional imaging considers multiple scattering as noise and intends to filter them out to obtain the ideal diffraction-limited image using the remaining ballistic waves. This strategy often fails when the ballistic signal is weaker than the multiple scattering recorded at the same time[3]. To the best of our knowledge, our proposed method is a first of its kind that enables the use of multiply scattered waves for microscopic image formation of an embedded object by computationally converting the waves to ballistic signal waves, which leads to a substantial increase in imaging depth. We demonstrated the tracing of 17 scattering events and enhanced the ballistic signal strength by almost 580 times. As a result, we could recover the object images, completely obscured in the conventional confocal imaging, with a microscopic spatial resolution better than 1.05 μm, even for in vivo through-skull imaging. Our proposed method works even for objects with extremely weak contrast such as myelinated axons and inhomogeneous tissue textures inside the brain. This is because the algorithm finds correlations among common object function contained in a large number of complex fields constituting the reflection matrix. The successful demonstration of our algorithm in this example supports that our work is well beyond the simple proof-of-concept level.

Our study marks an important milestone in solving high-order inverse scattering problems—considered the holy grail in the field of deep imaging[39]. It is worthwhile to emphasize that the proposed method works in the most general condition: an object is embedded within a thick scattering medium, and one can only access the back-scattered waves that undergo a roundtrip to the object. For this condition, there exists a vast amount of multiply scattered waves that do not interact with the object as well as those that reach the object. The experimental recording of a time-gated reflection matrix is the first critical step to detecting the multiple scattering with object information to the best possible degree while attenuating the unwanted multiple scattering. The proposed algorithm is a versatile and powerful technique that can selectively trace the multiple scattering events carrying the object information by exploiting their intrinsic wave

correlations in the recorded reflection matrix. The algorithm is particularly robust because it processes only the experimentally recorded reflection matrix to rectify the multiple scattering rather than compare the data with any theoretical model. Our approach is a general framework for solving the inverse problem of wave scattering. Therefore, it can also be applied to a wide range of wave imaging modalities, including ultrasonic imaging and microwave inspection[40,41]. The concept was demonstrated in microscopic imaging in the present study, but it can be extended to macroscopic imaging as long as wave properties are used for image formation.

The MST algorithm finds a set of phase plates that generate a similar transmission matrix to that of the scattering medium covering the target object. Notably, our study shows that the identified phase plates are the depth-sectioned transmission phase images of the real structures constituting the scattering medium. Essentially, our methods provide the 3D transmission phase map of the scattering medium itself as well as the reflectance image of the object. The transmittance phase images provide better contrast in visualizing cell bodies and obtaining their refractive index, but they could be obtained only for thin-section tissues in the transmission-mode microscope[42]. On the contrary, our approach finds them for a thick tissue in situ in the reflection-mode imaging. The benefit of our MST approach is not limited to object image reconstruction. One can physically fabricate a multi-layered inverse scattering block whose transmission matrix is the inverse of the identified transmission matrix[43]. By attaching the inverse scattering block to the surface of the scattering medium, optical clearing of the scattering medium can be realized without damaging the scattering medium itself. This can exempt the need for chemical processing in the tissue clearing[44]. In the context of deep-tissue optical imaging, one can find a specific excitation wavefront that can generate a sharp focus at a respective position in the object plane. By shaping the incident wave with a spatial light modulator, it will be possible to achieve a substantial depth increase in fluorescence imaging and super-resolution imaging[45], where a tight focusing of light or precise control of illumination patterns inside a scattering medium is a prerequisite. The key advance of this approach with respect to previous adaptive optics is to control substantial multiple scattering to form a focus.

Our algorithm finds multiple scattering trajectories based on the wave correlation of multiple scattering having interacted with a target object. This approach is intuitive, robust, and cost-effective. However, this doesn't trace all the multiple scattering containing the object information. One may consider combining the MST algorithm with other computational approaches such as compressive sensing and deep learning to increase the traceable multiple scattering trajectories[46,47]. Other approaches to finding the multiple phase masks or designed structures that produce a similar scattering matrix as the measured one could provide another possibility, although their scope is different from imaging[48,49]. Imaging geometry is another defining factor. It determines the lateral and axial resolutions of the layers that the algorithm can identify (see details in the Methods section). Consequently, it largely defines the maximum number of layers or the maximum number of scattering events that our algorithm can trace. Various collection geometries could be considered to better capture the multiple scattering of interest. The width of time gating is shortened as much as possible in conventional imaging to better rule out the multiple scattering, but there may be an optimal time gating window for collecting useful multiple scattering. The extension of the algorithm to incorporate backscattering in the scattering medium can be another important direction. Future studies addressing all these factors will extend the degree of multiple scattering coverage.

## Methods
### Experimental setup details
The laser scanning reflection-matrix microscopy technique[30,31] is based on an interferometric confocal reflectance microscope. A pulsed laser

 

(INSIGHT X3, Spectra physics, 1.3 μm wavelength, 19 nm bandwidth) was used as a broadband light source, providing a time-gating window of 25 μm in the optical path length. A pair of galvanometer mirrors were used for raster scanning a focused laser beam at the sample plane. The wave field backscattered from the sample was descanned and detected by a camera (InGaAs, Cheetah 800, Xenics, 6.8 kHz frame rate) based on the off-axis digital holographic imaging method. For interferometric detection, a separate planewave reference beam was combined at the camera in an off-axis configuration.

### Construction of a reflection matrix

We scanned a focused laser beam point-by-point with a half-wavelength scanning interval of $\Delta x_0$ over a certain ROI of size $L_0 \times L_0$. For all the $N_0 \times N_0$ sampling points with $N_0 = L_0/\Delta x_0$, we obtained $N_0^2$ electric-field images. Each image was laterally shifted according to the associated input scan position, and the area equal to the ROI on the shifted image was cropped to a size of $N_0 \times N_0$ pixels. Then, we obtained a set of $N_0^2$ field images, each of size $N_0 \times N_0$ pixels, in the laboratory frame. Finally, we constructed an $N_0^2 \times N_0^2$ reflection matrix $\boldsymbol{R}_{z_0,z_0}$ by mapping each 2D electric-field image into a column vector of $\boldsymbol{R}_{z_0,z_0}$ (see Supplementary Section Experimental setup for laser scanning reflection-matrix microscopy for details).

### Multiple scattering tracing algorithm

In MST algorithm, we first transform the measured matrix $\boldsymbol{R}_{z_0,z_0}$ to $\boldsymbol{R}_{z_N,z_N}$ and then transform $\boldsymbol{R}_{z_N,z_N}$ in such a way that the input plane is converted from the $N^{\text{th}}$ layer at $z_N$ to the $k^{\text{th}}$ layer at $z_k$, and the output plane is converted to the object layer at $z_0$. This can be implemented by multiplying the back-propagation operators $\boldsymbol{P}^*_{z_k,z_N}$ and $\boldsymbol{P}^*_{z_0,z_N}$ to the input and output sides of $\boldsymbol{R}$, respectively: $\boldsymbol{R}_{z_0,z_k} = \boldsymbol{P}^*_{z_0,z_N}\boldsymbol{R}_{z_N,z_N}\boldsymbol{P}^*_{z_k,z_N}$, where the superscript * denotes the complex conjugation.

Now, the transformed reflection matrix $\boldsymbol{R}_{z_0,z_k}$ can be decomposed into two terms as follows by expanding the scattering events in all layers except the one in the $z_k$ plane during the illumination process into the ballistic transmission and scattering components (see the detailed derivation in Supplementary Section Inverse scattering model):

$$\boldsymbol{R}_{z_0,z_k} = \boldsymbol{O}\boldsymbol{P}_{z_0,z_k}\boldsymbol{\Phi}_k + \boldsymbol{M} \qquad (2)$$

In this equation, the first term describes the contribution of the wave scattered only at the $k^{\text{th}}$ layer, where $\boldsymbol{\Phi}_k$ is a diagonal matrix whose elements are filled with $e^{i\varphi_k(\boldsymbol{\rho})}$. The second term, $\boldsymbol{M}$, accounts for the contribution of all the waves scattered at other layers. As we will

discuss in the following and demonstrate in Supplementary Section Numerical study, the influence of this factor gradually diminishes as the number of iterations increases.

The matrix element in the $i^{\text{th}}$ row and $j^{\text{th}}$ column of $\boldsymbol{R}_{z_0,z_k}$ represents the electric field $E(\boldsymbol{\rho}_i,z_0;\boldsymbol{\rho}_j,z_k)$ measured at a point $P(\boldsymbol{\rho}_i,z_0)$ for a unit-amplitude point source located at a point $P(\boldsymbol{\rho}_j,z_k)$. Based on the model represented by Eq. (2), the matrix element can be expressed as:

$$E\left(\boldsymbol{\rho}_i,z_0;\boldsymbol{\rho}_j,z_k\right) = O(\boldsymbol{\rho}_i,z_0)G\left(\boldsymbol{\rho}_i,z_0;\boldsymbol{\rho}_j,z_k\right)e^{i\varphi_k(\boldsymbol{\rho}_j)} + E_M\left(\boldsymbol{\rho}_i,z_0;\boldsymbol{\rho}_j,z_k\right)$$
$$(3)$$

Here, $G(\boldsymbol{\rho}_i,z_0;\boldsymbol{\rho}_j,z_k)$ is the free-space Green's function[50] of a single point source, which is a complex field of the spherical wave emanating from a point $P(\boldsymbol{\rho}_j,z_k)$, observed at a point $P(\boldsymbol{\rho}_i,z_0)$, and $E_M$ is the matrix elements of $\boldsymbol{M}$, representing the electric field of all the multiply scattered waves (see Supplementary Section Inverse scattering model for the detailed derivation of Eqs. (2) and (3)).

Next, we elaborate on the key concept to identify $\varphi_k(\boldsymbol{\rho}_j)$ from the measured reflection matrix. The first term on the right-hand side of Eq. (3) contains the object's amplitude reflectance function $O(\boldsymbol{\rho}_i,z_0)$ and the free-space Green's function. In Fig. 7a, b, we illustrate the first term in Eq. (3) of the two electric fields $E_1(\boldsymbol{\rho}_i,z_0;\boldsymbol{\rho}_1,z_k)$ and $E_2(\boldsymbol{\rho}_i,z_0;\boldsymbol{\rho}_2,z_k)$ for two different source positions $P_1(\boldsymbol{\rho}_1,z_k)$ and $P_2(\boldsymbol{\rho}_2,z_k)$, respectively. The two fields resemble each other, except for the center of the spherical phase profile and the overall phase retardation. After normalizing out the Green function in each field, we compute the field–field correlation to determine the relative phase retardations $\varphi_k(\boldsymbol{\rho}_2) - \varphi_k(\boldsymbol{\rho}_1)$ (Fig. 7b). The presence of $E_M$ with its magnitude much larger than the first term makes it difficult to find $\varphi_k(\boldsymbol{\rho}_j)$ accurately and gives rise to an error. As we shall explain below, the field correlation across the object plane raises the fidelity of finding $\varphi_k(\boldsymbol{\rho}_j)$, because the first term of Eq. (3) is coherently added with respect to the reference point at $\boldsymbol{\rho}_{\text{ref}}$, whereas $E_M$ is incoherent in the estimation of the correlation. Thus, the fidelity is determined by the number of detection pixels in the object plane.

The mathematical description of finding $\varphi_k(\boldsymbol{\rho})$ illustrated in Fig. 7 is given below. In Eq. (3), $O(\boldsymbol{\rho}_i,z_0)$ is independent of the illumination position $P(\boldsymbol{\rho}_j,z_k)$. Therefore, we can obtain the phase value $e^{i\varphi_k(\boldsymbol{\rho}_j)}$ of each illumination point by comparing the normalized electric fields by the Green's function (eliminating spherical wave curvature):

$$E_{\text{norm}}\left(\boldsymbol{\rho}_i,z_0;\boldsymbol{\rho}_j,z_k\right) = E\left(\boldsymbol{\rho}_i,z_0;\boldsymbol{\rho}_j,z_k\right)/G\left(\boldsymbol{\rho}_i,z_0;\boldsymbol{\rho}_j,z_k\right) \qquad (4)$$

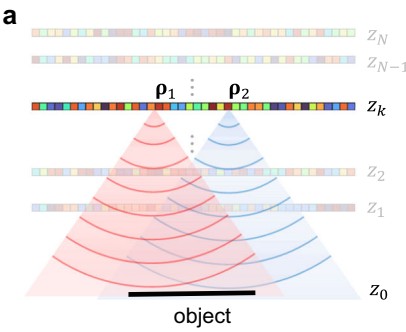
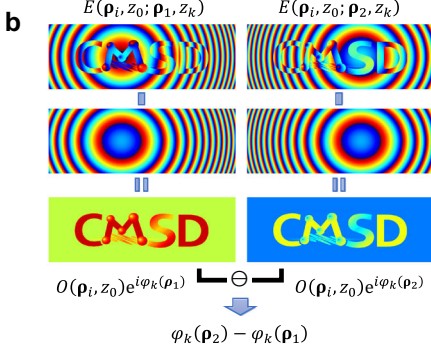

**Fig. 7 | Working principle of the quantification of $\varphi_k(\rho)$ in MST algorithm.** **a** Illustration of the first term of $E(\boldsymbol{\rho}_i,z_0;\boldsymbol{\rho}_j,z_k)$ in Eq. (3), which is the $j^{\text{th}}$ column vector of the propagated reflection matrix $\boldsymbol{R}_{z_0,z_k}$ with $j = 1,2$. **b** Illustration of phase correlation method for finding the phase functions $\varphi_k(\boldsymbol{\rho}_j)$. Examples of $E(\boldsymbol{\rho}_i,z_0;\boldsymbol{\rho}_1,z_k)$ and $E(\boldsymbol{\rho}_i,z_0;\boldsymbol{\rho}_2,z_k)$ are shown at the first row. After normalizing the

respective Green's functions (second row), only the object function remains along with $\varphi_k(\boldsymbol{\rho}_1)$ and $\varphi_k(\boldsymbol{\rho}_2)$. Based on the correlation of the two field images in the third row, an approximate phase function $\varphi_k(\boldsymbol{\rho}_2) - \varphi_k(\boldsymbol{\rho}_1)$ can be obtained. All the images in **d** represent phase maps. Note that the illustrations in **a** and **b** show only the first term in Eq. (3) for clarity.

The inner product of the $j^{th}$ and $l^{th}$ column vectors after normalizing the associated Green's function can be written as

$$\sum_i E_{\text{norm}}\left(\boldsymbol{\rho}_i, z_0; \boldsymbol{\rho}_j, z_k\right) \cdot E^*_{\text{norm}}\left(\boldsymbol{\rho}_i, z_0; \boldsymbol{\rho}_l, z_k\right) = e^{i[\varphi_k(\boldsymbol{\rho}_j) - \varphi_k(\boldsymbol{\rho}_l)]} \sum_i |O(\boldsymbol{\rho}_i, z_0)|^2 + M$$

(5)

Here, $M$ represents the contribution of multiple scattering caused by the other layers. As we will discuss in the following and demonstrate in Supplementary Section Numerical study, the influence of this factor gradually diminishes as the number of iterations increases.

Without loss of generality, we consider $\boldsymbol{\rho}_l = 0$ as a reference point and set the phase angle at the reference point $\varphi_k(\boldsymbol{\rho}_l = 0) = 0$. We can then obtain the first-round estimation of the phase function $\varphi_k$ from Eq. (5):

$$\varphi_k^{(1)}\left(\boldsymbol{\rho}_j\right) = \arg\left\{\sum_i E_{\text{norm}}\left(\boldsymbol{\rho}_i, z_0; \boldsymbol{\rho}_j, z_k\right) \cdot E^*_{\text{norm}}\left(\boldsymbol{\rho}_i, z_0; 0, z_k\right)\right\} = \varphi_k\left(\boldsymbol{\rho}_j\right) + \delta\varphi_k^{(1)}\left(\boldsymbol{\rho}_j\right)$$

(6)

Although we described the phase quantification process in Eq. (6) as if we compare only two column vectors of propagated reflection matrix $\boldsymbol{R}_{z_0, z_k}$, the actual algorithm used in our study utilizes the whole column vectors to determine the phase function with substantially enhanced the fidelity of convergence. More specifically, we define a Green's function normalized matrix $\boldsymbol{S}_{z_0, z_k}$ whose matrix elements are given by

$$\boldsymbol{S}_{z_0, z_k}\left(\boldsymbol{\rho}, z_0; \boldsymbol{\rho}', z_k\right) \equiv \frac{E\left(\boldsymbol{\rho}, z_0; \boldsymbol{\rho}', z_k\right)}{G\left(\boldsymbol{\rho} - \boldsymbol{\rho}'; z_0 - z_k\right)} = e^{i\varphi_k(\boldsymbol{\rho}')}O\left(\boldsymbol{\rho}, z_0\right) + \boldsymbol{M}'\left(\boldsymbol{\rho}; \boldsymbol{\rho}'\right)$$

(7)

Then we quantify the phase function of $k^{th}$ phase plate by solving

$$\arg\min_{\boldsymbol{\varphi}_k^{(1)}}||\boldsymbol{S}_{z_0, z_k} - \tau\boldsymbol{o}^T \times e^{i\boldsymbol{\varphi}_k^{(1)}}||_2$$

(8)

where $\boldsymbol{o}^T$ is a transpose of the vectorized object function (column vector), $e^{i\boldsymbol{\varphi}_k^{(1)}}$ is vectorized phase function of the corresponding phase plate (raw vector), and $||\boldsymbol{A}||_2$ implies the Frobenius norm of a matrix $\boldsymbol{A}$. This minimization process is identical to finding the first singular vector of $\boldsymbol{S}_{z_0, z_k}$, and we utilized the well-known power iteration method[51].

The phase angle $\varphi_k^{(1)}(\boldsymbol{\rho}_j)$ obtained by Eqs. (6–8) is asymptotically close to the ground-truth phase angle $\varphi_k(\boldsymbol{\rho}_j)$ with the error $\delta\varphi_k^{(1)}(\boldsymbol{\rho}_j)$ owing to $M$. In general, the phase error $\delta\varphi_k^{(1)}$ is not negligible owing to the multiple scattering by the other layers. For this reason, we used a superscript (1) to indicate that Eqs. (6–8) provides the first guess. Even if $\varphi_k^{(1)}(\boldsymbol{\rho}_j)$ is not exact, multiplying $e^{-i\varphi_k^{(1)}(\boldsymbol{\rho}_j)}$ to the right side of $\boldsymbol{R}_{z_0, z_k}$ compensates $\varphi_k(\boldsymbol{\rho}_j)$ to some extent. In effect, this is equivalent to placing a compensating phase plate of $-\varphi_k^{(1)}(\boldsymbol{\rho}_j)$ adjacent to the $k^{th}$ phase layer that it reduces the scattering induced by the $k^{th}$ layer at $z_k$. After correcting $\varphi_k^{(1)}(\boldsymbol{\rho}_j)$, we move to next layer at $z_q(q \neq k)$ by multiplying the corresponding propagation matrix, and then perform the same procedure. To account for the scattering layers in the output pathway, i.e. $\boldsymbol{T}^T$ in $\boldsymbol{R}_{z_N, z_N}$ in Eq. (1), we take the transpose of $\boldsymbol{R}_{z_N, z_N}$ and repeat the same steps of corrections. Since this first round only partially corrects the multiple scattering, we iterate this cycle of operations until the phase retardations at the $n^{th}$ cycle, $\varphi_k^{(n)}$, converge with a certain tolerance. The summation of the phase functions over the iterations, $\varphi_k^c(\boldsymbol{\rho}_j) = \sum_n \varphi_k^{(n)}(\boldsymbol{\rho}_j, z_k)$ converges to $\varphi_k(\boldsymbol{\rho}_j, z_k)$, which results in the finding of $\boldsymbol{T}$ and the object function $O(\boldsymbol{\rho}_i, z_0)$ (see Supplementary Section Inverse scattering model for the detailed iteration process).

## Sampling considerations

**Area of the phase function recovered by the MST algorithm.** The MST algorithm intends to quantify the phase functions $\varphi_k(\boldsymbol{\rho}_j)$ for multiple discrete layers. Due to the imaging geometry, the diameter $L_k$ of the circular FOV for the phase function $\varphi_k$ is finite. $L_k$ is determined by the distance from the object plane to the scattering layer $z_k - z_0$, the numerical aperture of the objective lens, and the ROI size $L_0 \times L_0$ at the object plane. $L_k$ is approximately given by $L_k \simeq 1.2z_k + L_0$ under our experimental condition (see details in Supplementary Section Sampling area of the phase functions).

**Lateral and axial resolutions of mapping the phase function.** The square-shaped ROI of size $L_0 \times L_0$ at $z_0$ serves as an aperture in the mapping of $\varphi_k$ (Eqs. (4–6)). This means that the effective numerical aperture of quantifying objects in $z_k$ plane is given by $L_0/(2z_k)$. Therefore, the lateral spatial resolution $\Delta x_k$ of identifying $\varphi_k$ is proportional to the inverse of the numerical aperture, $2z_k/L_0$. Likewise, the axial resolution $\Delta z_k$ of quantifying $\varphi_k$ also increases inversely with respect to the square of the effective numerical aperture, $(2z_k/L_0)^2$. The MST algorithm can recover scatters equal to or larger than the spatial resolution $\Delta x_k$ and $\Delta z_k$ of mapping $\varphi_k$ (see details in Supplementary Section Spatial resolution of identifying phase function φk).

**Computational cost of the MST algorithm.** The computation time of the MST algorithm is mainly determined by the matrix multiplications for getting access to individual layers. The size of the reflection matrix $\boldsymbol{R}$ and the propagation matrix $\boldsymbol{P}$ for a square-shaped ROI with $L_0 = N_0\Delta x_0$ is $N_0^2 \times N_0^2$, where $\Delta x_0$ is lateral resolution in the object plane at $z_0$. Then, the computational time for the matrix multiplication $\boldsymbol{RP}$ is proportional to $N_0^6 = (L_0/\Delta x_0)^6$. To accelerate the speed of MST algorithm, we make use of a graphic processing unit (GPU, Nvidia RTX A6000) for the matrix multiplication. With a custom-made software based on the Matlab, the computational time for the data in Fig. 3 ($N_0 = 100$) was about 500 s, while that for the data in Figs. 4 and 5 ($N_0 = 160$) was about 3.4 hours (see more discussions in Supplementary Section Computational cost). The number of phase plates can be increased up to the point that their spacings correspond to the achievable axial resolution set by the imaging geometry. Since the computational cost for increasing the number of phase plates is substantial, we choose a suitable condition where the gain in the ballistic signal enhancement is saturated. It is noteworthy that the application of the MST algorithm is post-processing after acquiring the reflection matrix. Therefore, the heavy computational cost of the MST algorithm does not affect in vivo applications which require fast reflection matrix acquisition speed. If necessary, we can speed up the application of the MST algorithm by dividing the ROI into multiple patches with smaller $N_0$. For example, the computation time for the data presented in Figs. 4, and 5 can be 80 times faster with $3 \times 3$ patches of $N_0 = 54$.

## Animal preparation with intact skull window

All animal procedures were approved by the Korea University Institutional Animal Care and Use Committee (KUIACUC-2022-0013) and conducted in compliance with the ethical standards of KUIACUC. In this study, we used 20-week-old C57BL/6 mice. The animal preparations with an intact skull window were performed following the previous reported procedure[30]. The mice were anesthetized with 1.5–2% isoflurane (to maintain a breathing frequency of around 1 Hz). Their bodies were warmed at 37–38 °C by a heat blanket, and the eyes were covered with an eye ointment during the surgery and imaging. Further, the hairs were removed with a Nair hair remover, and a midline scalp incision was performed to expose both the parietal plates of the skull. After sterile saline was applied to the skull, the connective tissue

remaining on the skull was gently removed with the sterilized forceps. The scalp covering the skull was removed completely and then a sterile coverslip of 5-mm diameter (#1, round shape, Warner Instruments, USA) was attached to the center of the parietal bone using an ultraviolet-curable glue. Finally, a custom-made metal plate was attached to the skull with cyanoacrylate to keep the mouse's head immobilized during the imaging. For the imaging, the mice were anesthetized with isoflurane (1.3–1.5% in oxygen to maintain a breathing frequency of around 1.5–2 Hz) and placed on a three-dimensional motorized stage heated by a heat blanket at 37–38 °C.

## Reporting summary

Further information on research design is available in the Nature Portfolio Reporting Summary linked to this article.

## Data availability

The datasets acquired for this study are available from the corresponding authors upon request. Due to the substantial size of our imaging data (approximately 400GB), it presents challenges for public repository uploads.

## Code availability

The code implemented in this study is available from the authors upon request.

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

## Acknowledgements
This work was supported by the Institute for Basic Science (IBS-R023-D1) (S.K., Y.K., H.L., S.K.*, J.H.H., S.Y., and W.C.). This work was supported by the National Research Foundation of Korea (NRF) grant funded by the Korea government (MSIT) (No. RS-2023-00213310) (S.Y.).

## Author contributions
S.Y., S.K., and W.C. conceived the project, S.Y. and S.K. designed the MST algorithm along with H.L. S.K. and S.Y. implemented the algorithm, and S.K. conducted the data analysis including numerical simulation and experimental data with S.Y. Y.K., S.K., S.Y., and S.K.* took the experimental data, and J-.H.H. prepared biological samples. S.K., S.Y., and W.C. prepared the manuscript, and all authors contributed to finalizing the manuscript. W.C. supervised the project. S.K. and S.K.* indicate Sungsam Kang and Seho Kim, respectively.

## Competing interests
The authors declare no competing interests.
