## [Peer Review File · Nature Communications]

Tracing multiple scattering trajectories for deep optical imaging in scattering mediaReviewer #1 (Remarks to the Author):

The authors demonstrate a novel optical technique to image through scattering media based on an experimentally measured time-gated reflection matrix.

Their computational analysis of this matrix breaks down the complex thick scattering layer into several phase plates 'separated' by free space, which can then be individually compensated by numerically adding the conjugate phases. One can then extract from the computationally corrected reflection matrix a confocal reflectance image of the sample located below the thick scattering medium.

While it is based on a previously described experimental measurement, the analysis method is original and yield some very convincing results. In particular, the authors were able to demonstrate this in living mice, by imaging axons through the intact skull. This technique is restricted to forward scattering media, but most biological tissue share this property. A significant advantage as compared to previous work from the same group (ref. 29) is that no active element is needed to perform the correction, as it only relies on the reflectance matrix post-processing.

While this technique is still limited to reflectance contrast, which may lack specificity, it represents a significant step in deep tissue imaging, and is likely to be of great interest for the community and stimulate further works in the field.

I therefore recommend for publication, once the authors addressed the following remarks. (In the following, lines are numbered continuously using the Word option in tab Layout, menu Line numbers.)

line 17: 'our method rectifies multiple scattering'
Please provide corresponding optical depth.

50 - 53: The definition of what is a 2nd or a 3rd order approach should be more explicit.

70: '17 scattering events'
Please define more specifically what is a scattering event here. It is rather clear once we read the paper, but a bit confusing here. A clarification of its relationship to the scattering mean free path could also be beneficial.

182: The introduction of polygonal and number shapes in the phase patterns in Fig.3 affects quite a bit the statistical distribution of typical complex speckle fields and the associated phase patterns. Does this influence the convergence of the proposed reconstruction algorithms?
I understand the handiness of such patterns, but they are probably not so essential as one can easily quantify the success of the reconstruction with such synthetic data.

220-221: how are optical thicknesses computed?

236-249: why mention $x_{s_i_2}$ then? This section is a bit confusing.
This factor is used then when analysing results in Fig.5, but there is not clear justification then on why this factor is used as compared to $x_{s_i_1}$. Please clarify this.

277, Fig.4b: To me, there is still quite some ballistic components as we can distinguish pretty clearly the shape of the target. It is same for Fig.5b. Given the high performances of the proposed algorithm, it would be very interesting to see how it evolves at larger optical thicknesses. If there is some limit that prevent from doing so, please provide some comments on this.

342: typo? 'pate pates'

406: it could be useful to specify that this technique is *numerically* converting multiply scattered waves into ballistic ones.

580: R_{z_0, z_0} is the measured matrix in section 1.2 from supplementary materials, as well in the 'construction of a reflection matrix' part of the Methods section in the main text.

But here R_{zN}, zN is said to be the measured one. Please clarify this in all relevant sections.

584: typo ? R_{z0}, zn should be R_{zo}, zk

586: in Eq.(2), it is not so trivial that M can be simply added like this. The authors should explicitly refer to Eq.(S12), and give some quick insights on how this can be interpreted. Matrix ϕ is only introduced in supplementary material but not here in the main text.

607: in Eq.(5) as well as in Eq.(S12), the M terms still depend on O , ϕ_k , and E . Please justify more rigorously why they can still be isolated.

614: Eq.(6) and Eq.(S15) are actually valid on the support of object O . If the object is scarce as for the myelinated axons, how does it affect the convergence of the phase correction ?

673-674: typo? 'The saline covering the skull'
I guess the authors meant the *scalp* covering the skull.

677: typo 'to keep keeping'

supp inf:
section 2.2: MW is not defined after Eq.(S12).

Reviewer #2 (Remarks to the Author):

Report on "Tracing multiple scattering trajectories for deep optical imaging in scattering media"

This article deals with imaging through disordered media that scatter light multiple times. Such distortions due to wave scattering constitute a major bottleneck for bio-medical imaging as traditional approaches typically fail in this context. Here, an approach is proposed and implemented that mimics a scattering medium through multiple complex phase plates, whose phase distortions are found computationally (in post-processing of reflection matrix measurements).

I find this work conceptually appealing and of interest to the community working on multiple scattering, inverse problems and imaging techniques. There are, however, also several weak points that need to be addressed by the authors:

◦ The idea to compensate multiple forward-scattering through a cascade of phase masks is not new (see arXiv:2204.02865). Also the compensation of multiple scattering (both in forward and backward direction) through a physically fabricated anti-reflection structure has recently been presented (see Nature 607, 281 (2022)). With the first work being numerical only and the second one being implemented in the microwave domain, I still see room for the present work in the literature, but it clearly needs to be better embedded in the state of the art. My impression is that also recent work by Aubry et al. (arXiv:2303.06119 & arXiv:2303.07483) has a certain closeness to the current manuscript that deserves to be discussed.

◦ What I clearly missed in the main text of the manuscript is an easily accessible explanation of how the MST algorithm works and how it is able to determine the required degrees of freedom in the phase masks. (I also found the explanation in the methods section hard to follow.) Maybe it would also be helpful in this context to include an explanatory figure that illustrates the basic principles of the algorithm. Also the new method's limitations should be more clearly spelled out in terms of real-world applicability.

◦ I stumbled over the following sentence: "In the first strategy, we measure the electric fields of the backscattered waves, including both the ballistic wave and multiply scattered waves, whose flight times are identical to that of the ballistic wave." How can it be that a wave that is scattered can have the same flight time as a ballistic wave, especially when it also needs to reach the

target? Doesn't scattering necessarily lead to an increase in the flight time as compared to the straight-line path of a ballistic wave?

° Which restrictions do apply to the target? The authors use a specific target here with a high contrast between bright and dark parts. Would their approach also be applicable to objects with a weak contrast?

° It isn't clear to me how the authors determine the number of scattering events that they can compensate for (like 17 in the conclusion section).

° It may also require further explanation as to why the method can ultimately be applied to systems with very pronounced wave backscattering (such as the mouse skull).

° In Eq. 6 the "arg" seems to appear twice.

In summary, this is a very interesting contribution to the emerging field of matrix imaging. After the authors have had a chance to address the points raised above, I would be glad to have another look at the manuscript.

Reviewer #1

The authors demonstrate a novel optical technique to image through scattering media based on an experimentally measured time-gated reflection matrix. Their computational analysis of this matrix breaks down the complex thick scattering layer into several phase plates 'separated' by free space, which can then be individually compensated by numerically adding the conjugate phases. One can then extract from the computationally corrected reflection matrix a confocal reflectance image of the sample located below the thick scattering medium.

While it is based on a previously described experimental measurement, the analysis method is original and yields some very convincing results. In particular, the authors were able to demonstrate this in living mice, by imaging axons through the intact skull. This technique is restricted to forward scattering media, but most biological tissues share this property. A significant advantage as compared to previous work from the same group (ref. 29) is that no active element is needed to perform the correction, as it only relies on the reflectance matrix post-processing.

While this technique is still limited to reflectance contrast, which may lack specificity, it represents a significant step in deep tissue imaging, and is likely to be of great interest for the community and stimulate further works in the field. I therefore recommend for publication, once the authors addressed the following remarks. (In the following, lines are numbered continuously using the Word option in tab Layout, menu Line numbers.)

We are deeply grateful to the reviewer for recognizing the importance of our research. This study introduces a technique that addresses high-order inverse scattering problems under the broadest and most practical conditions, including *in vivo* deep-tissue imaging in reflection geometry. Fundamentally, this approach enables us to utilize multiple light scattering in a deterministic manner to reconstruct microscopic images. As noted by the reviewer, we believe this is a significant advance in the field of deep tissue imaging, opening new possibilities and encouraging further research in this domain.

We also thank the reviewer for the thorough examination of our paper and for insightful comments and suggestions. In this response letter, we have addressed each question, and we are confident that this has substantially improved both scientific integrity and readability.

To begin with, we'd like to provide clarifications on the reviewer's comment about our study being limited to reflectance contrast. While we agree that reflectance imaging has less specificity compared to fluorescence imaging, its ability to visualize myelinated axons and blood vessels is nonetheless significant. Moreover, when considering human subjects instead of animal studies, the fact that no labeling is required becomes a crucial advantage and may lead to important tools in future medical applications.

However, there is something even more important. Whether we perform reflectance imaging, fluorescence imaging, or any other type of microscopic imaging, we must eventually decipher how the scattering medium surrounding the object of interest distorts light propagation. In most cases, this propagation is determined by elastic scattering. Our measurement of the reflection matrix and the subsequent solving of inverse problems aimed at tracing the elastic scattering events caused by multiple light scattering, and the resulting product is reflectance imaging. Once we understand the characteristics of light propagation in a scattering medium *in situ*, we can use the information to enhance the imaging depth of any microscopic imaging modality. We can determine which light pattern will allow us to focus light on the desired position through the identified layers within the scattering medium. This knowledge can be utilized for imaging with high-specificity modalities, such as fluorescence imaging. Examples include our recent work on deep-tissue single-molecule localization microscopy (Park, S. et al. Label-free adaptive optics single-molecule localization microscopy for whole zebrafish. Nat. Commun. 14, 4185 (2023)). From this perspective, the proposed method will make a truly broad impact on the field of deep-tissue imaging.

line 17: 'our method rectifies multiple scattering' Please provide corresponding optical depth.

Our method transforms multiple scattering noises into ballistic signals, thereby amplifying the ballistic signals by almost 600 times. The optical path is dictated by the attenuation level of the ballistic wave, as represented by the formula $I_b = I_0 e^{-2z/l_s}$. Here '2' factor in the exponent accounts for the roundtrip in the reflection geometry of detection. If we introduce η as the enhancement factor for the intensity of the ballistic wave through the rectification of multiple scattering, the optical depth after rectification can be represented as $I'_b = \eta I_b = I_0 e^{-2z'/l_s}$. This equation implies that the optical path length is decreased by $\Delta z = z - z' = \left(\frac{l_s}{2}\right) \ln(\eta)$, indicating that the scattering medium is less scattering. For an enhancement factor η of 600, Δz equals $3.2 l_s$, suggesting that our method extends the imaging depth by $\Delta z = 3.2 l_s$.

In simpler terms, our method transforms multiple scattering noises into useful signals, thereby enabling us to see deeper into the sample than would otherwise be possible. The increase in imaging depth is determined by the magnitude of the enhancement factor, with a factor of 600 leading to a considerable extension of the imaging depth.

Pursuing the reviewer's recommendation, we have incorporated the following sentences to detail the enhancement of imaging depth in terms of the optical depth:

“By implementing the inverse scattering using the identified phase plates, our method rectifies multiple scattering and amplifies ballistic waves by almost 600 times. This leads to a significant increase in imaging depth - more than three times the scattering mean free path - as well as the correction of image distortions.”

We also added the following sentences to the end of the introduction section:

“This effectively reduces the optical thickness of the scattering medium by more than three times the scattering mean free path. As a result, the imaging depth increases by the same factor of the scattering mean free paths.”

50 - 53: *The definition of what is a 2nd or a 3rd order approach should be more explicit.*

The classification of inverse scattering problems is based on the count of unknown layers required to be identified. To further clarify the classification of inverse scattering problems, we have included the following explanation:

“In this context, imaging modalities can be classified by the number of scattering events that they can trace, which corresponds to the order of Born approximation they attempt to address (Ref: M. Born and E. Wolf, Principles of Optics, 7th ed. Cambridge Univ. Press, 2003, ch. Scattering from inhomogeneous media, pp. 695–734.). This classification essentially involves determining the number of unidentified layers the modalities seek to map.”

“For instance, adaptive optics, which deals with the perturbation of either the excitation or returning beams, can be viewed as second- or third-order methods. This classification is due to their attempt to identify the input/output pupil layers in addition to a sample layer.”

70: *'17 scattering events' Please define more specifically what is a scattering event here. It is rather clear once we read the paper, but a bit confusing here. A clarification of its relationship to the scattering mean free path could also be beneficial.*

In response to the reviewer's comment about clarifying what constitutes a 'scattering event' and its relationship with the scattering mean free path, we provide the following explanation: In the MST model, each phase plate induces either a single scattering or no scattering at all. For instance, if a phase plate is akin to a thin diffraction grating, the first-order diffraction corresponds to a single scattering, and the zeroth-order diffraction corresponds to no interaction. Consequently, when modeling a scattering medium with 8 discrete layers, up to 17 scattering events can occur. These events account for wave propagation on the way in and out, and the reflection by the target object.

However, this does not mean that the MST algorithm cancels a scattering medium with a thickness of up to $17 l_s$. The scattering mean free path (MFP) is the average distance between successive scattering events. At each layer, there's a certain probability of scattering occurring or not. Therefore, despite a maximum of 17 scattering events, the average number of scattering events could be significantly fewer. For instance, the derived phase plates from mouse cranial bone shown in Fig. 5 correspond to an average scattering length of $6.4 l_s$ in the roundtrip, which is much less than 17. It's important to note that the rectification of the multiple light scattering events, based on these phase plates, reduces the optical thickness by the equivalent number of MFPs.

We revised the corresponding text of the main manuscript (line 70) to make it clear what the term 'scattering event' indicates.

“With the MST algorithm, we could identify up to, but not limited to, eight layers of phase plates representing thick skull tissue and a layer of underlying target biological structures. Considering that each phase plate and the target structure can induce scattering events, this corresponds to tracing a total of 17 scattering events—eight on the way in, one induced by the target structure, and another eight on the way out.”

We also added a more explicit explanation of the relation between the number of traced scattering events and the scattering mean free path in the section, 'Demonstration of MST algorithm with a highly scattering skull tissue.'

“Let us elaborate on the relation between the number of traced scattering events and the scattering mean free path. The scattering mean free path is the average distance between successive scattering events. At each phase plate, there's a certain probability of scattering occurring or not. Therefore, despite a maximum of 17 scattering events, the average number of scattering events could be significantly fewer. For instance, the derived phase plates from mouse cranial tissue shown in Fig. 5 correspond to an average scattering length of $6.4 l_s$ in the roundtrip, which is much less than 17. The rectification of the multiple light scattering events, based on these phase plates, reduces the optical thickness by the equivalent number of the scattering mean free paths.”

182: The introduction of polygonal and number shapes in the phase patterns in Fig. 3 affects quite a bit the statistical distribution of typical complex speckle fields and the associated phase patterns. Does this influence the convergence of the proposed reconstruction algorithms? I understand the handiness of such patterns, but they are probably not so essential as one can easily quantify the success of the reconstruction with such synthetic data.

As rightly inferred by the reviewer, the introduction of polygonal and number shapes in the phase patterns was primarily for the ease of visual verification of the MST algorithm's effectiveness. In fact, we overlaid these artificial shapes with random speckle patterns, as depicted in Fig. 3b of the main manuscript, to mimic a realistic scattering medium. While not immediately apparent, the MST algorithm also managed to accurately recover the superimposed random patterns.

To further emphasize that our algorithm performs efficiently with random phase patterns, we conducted an additional numerical simulation. This simulation was executed with all phase plates filled exclusively with random phase patterns. As demonstrated in Fig. R1, the MST algorithm successfully reconstructed these general random phase structures with high fidelity. We quantified the accuracy of the MST algorithm's performance by computing the Pearson correlation coefficients (PCC) between the reconstructed phase map and the corresponding ground-truth values. The obtained correlation coefficients for the four layers were as follows: 0.87, 0.62, 0.64, and 0.71. Additionally, the PCC value for the reconstructed image of the target object and its ground truth reached 0.94 due to additional suppression of residual multiple scattering noise using confocal gating. This analysis confirmed the algorithm's robustness and accuracy in reconstructing random phase structures.

Figure R1. Numerical demonstration of MST algorithm with randomly patterned layers. **a-b**, Intensity map of confocal and MST images of the target, respectively, reconstructed from numerically generated data. Scale bar, 10 μm . Color scales for both images are normalized by the maximum intensity of **b**. **c**, Ground-truth phase map of phase plates used in the numerical simulation. Scale bar, 50 μm . **d-e**, Reconstructed phase maps from MST algorithm, and residual phase maps with respect to the ground-truth phase maps in **c**, respectively. Circles correspond to the reconstruction area in the MST algorithm whose diameter depends on the distance of the layer from the target object.

We added this new result to the Supplementary Section 3.3 and referred to it in the main text as follows.

“In addition, different polygonal patterns and numbers are superimposed on each layer for ease of performance evaluation of the proposed algorithm (see Supplementary Section 3.3 for the case of phase plates filled only with random phase patterns).”

220-221: how are optical thicknesses computed?

Ballistic waves, by definition, refer to wave components that maintain their propagation angles unchanged while propagating through a scattering medium. In this context, the diagonal elements of $\tilde{\mathbf{T}}$, the k-space transmission matrix of the scattering medium, correspond to the ballistic wave components of the medium. Consequently, the optical thickness d_{opt} of the numerically generated scattering medium in Fig. 3 can be determined by calculating the intensity ratio of the diagonal elements to the total intensity of the ground-truth transmission matrix $\tilde{\mathbf{T}}$ in the spatial frequency domain: $d_{\text{opt}}/l_s = -\log\left(\frac{\sum_i |\tilde{\mathbf{T}}_{ii}|^2}{\sum_i \sum_j |\tilde{\mathbf{T}}_{ij}|^2}\right) \sim 3.3$. Here $\tilde{\mathbf{T}}_{ij}$ indicates the matrix element. Similarly, the optical thickness d_{opt}^c after applying the inverse of quantified transmission matrix \mathbf{T}^c can be calculated as, $d_{\text{opt}}/l_s = -\log\left(\frac{\sum_i |\tilde{\mathbf{T}}'_{ii}|^2}{\sum_i \sum_j |\tilde{\mathbf{T}}'_{ij}|^2}\right)$ with $\tilde{\mathbf{T}}' = \tilde{\mathbf{T}}(\tilde{\mathbf{T}}^c)^{-1}$.

We added the following sentence to clarify how the scattering mean free path was estimated.

“Here the scattering mean free path was estimated by the intensity ratio between the diagonal elements and the total intensity in the spatial frequency domain transmission matrix.”

236-249: why mention x_{si_2} then? This section is a bit confusing. This factor is used then when

analysing results in Fig. 5, but there is not clear justification then on why this factor is used as compared to xsi_1. Please clarify this.

While ξ_1 estimates the accuracy of the quantified transmission matrix T^c , ξ_2 estimates the contrast of the confocal image before and after the application of MST algorithm. Specifically, it is defined by the enhancement of the diagonal elements in the space domain reflection matrix after the multiple scattering rectification. Our MST algorithm works in the direction of increasing ξ_2 by rectifying the multiple scattering noise (off-diagonal elements) into ballistic waves (diagonal elements). Essentially, ξ_2 assesses the final outcome of the MST algorithm, which is closely related to image metrics used in adaptive optics.

To clarify the reason why we introduced the quantity ξ_2 , we revised the related sentences as follows.

“Multiple scattering rectifications increase the ballistic signals and attenuate the multiple scattering background signals. This jointly enhances the signal-to-background ratio of the confocal image after the application of MST algorithm, and thus, the achievable imaging depth. Here, we introduce another criterion η_c that quantifies the ballistic signal enhancement in the reconstructed MST image: $\eta_c = \sum_i |R_{ii}^c|^2 / (\sum_i |R_{ii}|^2)$. Here, R_{ii} and R_{ii}^c are the diagonal elements of \mathbf{R}_{z_0, z_0} and \mathbf{R}_{z_0, z_0}^c , respectively.”

“ ξ_2 provides the direct measure of the enhancement of image quality, but there is an ambiguity in estimating α , especially when the initial ballistic signal is extremely weak. In this case, ξ_1 is a more reliable measure of ballistic signal enhancement than ξ_2 .”

277, Fig.4b: To me, there is still quite some ballistic components as we can distinguish pretty clearly the shape of the target. It is same for Fig.5b. Given the high performances of the proposed algorithm, it would be very interesting to see how it evolves at larger optical thicknesses. If there is some limit that prevent from doing so, please provide some comments on this.

As pointed out by the reviewer, the presence of the remaining ballistic components enables a partial recognition of the target's features, as depicted in Fig. 4b and Fig. 5b. However, it's important to bear in mind that these figures only visualize large-scale features. In fact, these figures are time-gated confocal reflectance images reconstructed from the measured reflection matrix. A significant portion of the multiple scattering components has already been excluded by the coherence gating and the confocal gating, allowing the preservation of a small portion of ballistic components despite the strong multiple scattering background.

As we discuss in Supplementary Section 3.2, the remaining ballistic components can be characterized from the average point spread function (PSF). We provide the PSFs of the experimental results in Figs. 4-6 in Fig. R2. As we move from Fig. 4 to 6, the severity of multiple scattering increases. As observed from the cross-sectional profile in Fig. R2c, a double-peak structure emerges with a narrow central ballistic peak and a broad multiple-scattering background. We define the relative contribution of the ballistic signal to the total peak of the PSF as the α parameter, and we have summarized this for each dataset in Table S1 of Supplementary Section 5.

As the reviewer pointed out, as the optical thickness increases and the ratio of ballistic-to-multiple scattering decreases, the ballistic peak becomes attenuated (Fig. R2f) and ultimately disappears (Fig. R2i). However, even when the ballistic peak is entirely obscured, our MST algorithm manages to convert multiple scattering into the signal, thus recovering a sharp PSF (Fig. S10c). Consequently, we could successfully reconstruct myelinated axons beneath a thick skull with near-diffraction-limited spatial resolution. This outcome underscores both the novelty and efficacy of our method.

Figure R2. Point spread functions of experimental data in Figs. 4-6. **a-c**, Confocal reflectance image, point spread function, and its cross-section along the x-axis, respectively, for the data shown in Fig. 4 (a resolution target under an onion tissue). Scale bar in **a**: 20 μm . Scale bar in **b**: 10 μm . **d-f**, Same as **a-c**, but for the data in Fig. 5 (a resolution target under a 180- μm -thick mouse skull). **g-i**, Same as **a-c**, but for the data in Fig. 6 (*in vivo* brain tissue imaging through a 200- μm -thick mouse skull).

We added this PSF analysis of the conventional optical coherence microscopy to Supplementary Section 5. We also added the following sentence to explain the reason why large features remain visible in conventional optical coherence imaging and referred readers to the PSF analysis in Supplementary Section 5.

“Fine microscopic structures were completely obscured while large features remained visible to some extent due to the residual ballistic signals (see Supplementary Section 5 for detailed PSF analysis).”

342: typo? 'pate pates'

Revise accordingly.

406: it could be useful to specify that this technique is **numerically* converting multiply scattered waves into ballistic ones.*

Revise accordingly.

580: R_{z_0, z_0} is the measured matrix in section 1.2 from supplementary materials, as well in the 'construction of a reflection matrix' part of the Methods section in the main text. But here R_{z_N, z_N} is said to be the measured one. Please clarify this in all relevant sections.

We appreciate the reviewer's careful reading of our manuscript. Indeed, R_{z_0, z_0} is the actual measured matrix in the experiment. R_{z_N, z_N} was introduced to explain the general concept of the reflection matrix (Fig. 2c and Fig. S3a) where the surface of the thick scattering sample is defined as the entrance and exit planes. To remove the ambiguity of the explanation, we revised the sentences as follows.

"In MST algorithm, we first transform the measured matrix R_{z_0, z_0} to R_{z_N, z_N} and then transform R_{z_N, z_N} in such a way that the input plane is converted from the N^{th} layer at z_N to the k^{th} layer at z_k , and the output plane is converted to the object layer at z_0 ."

584: typo ? R_{z_0, z_N} should be R_{z_0, z_k}

Revise accordingly.

586: in Eq.(2), it is not so trivial that M can be simply added like this. The authors should explicitly refer to Eq.(S12), and give some quick insights on how this can be interpreted. Matrix ϕ is only introduced in supplementary material but not here in the main text.

We appreciate the reviewer's comment and suggestion. We have revised the related paragraph to provide more insight into how Eq. (2) is derived and to guide readers to the supplementary material where the detailed derivation is provided:

"Now, the transformed reflection matrix R_{z_0, z_k} can be decomposed into two terms as follows by expanding the scattering events in all layers except the one in the z_k plane during the illumination process into the ballistic transmission and scattering components (see the detailed derivation in Supplementary Section 2.2):

$$R_{z_0, z_k} = OP_{z_0, z_k} \Phi_k + M. \quad (2)$$

In this equation, the first term describes the contribution of the wave scattered only at the k^{th} layer, where Φ_k is a diagonal matrix whose elements are filled with $e^{i\varphi_k(\rho)}$. The second term, M , accounts for the contribution of all the waves scattered at other layers."

607: in Eq.(5) as well as in Eq.(S12), the M terms still depend on O , ϕ_k , and E . Please justify more rigorously why they can still be isolated.

The first term in Eq. (S12) assumes that scattering events occur only once in k^{th} phase plate of the illumination process. In other words, wave components that undergo scattering by the other phase plates are excluded in the first term, and they are included in the second term, M . As the reviewer pointed out, M is also dependent on the object O and $\Phi_{k'}$ for $k \neq k'$. Consequently, as explained in Eq. (S15), the reconstructed k^{th} phase plate $\varphi_k^{(1)}$ in the initial step contains a phase error $\delta\varphi_k$ due to M .

However, the quantified phase plate $\varphi_k^{(1)}$ by Eq. (S14) is predominantly affected by the k^{th} phase plate because the contribution of other layers is averaged over the cone-shaped imaging geometry. As a result, the correction of the quantified phase map $\varphi_k^{(i)}$ at each i^{th} iteration step can gradually reduce the multiple scattering. In other words, as the progress of the iteration step, $\varphi_k^{(i)}$ are getting more and more accurate while the contribution of M is decreased.

Figure R2 shows the detailed process, reorganized from Fig. S8 in the supplementary information. In the initial iteration step, two representative images of $E(\rho, z_0; \rho_1, z_4)$, and $E(\rho, z_0; \rho_2, z_4)$ do not show

the detailed structure of the target because the contribution of \mathbf{M} is substantial. As a result, the quantified phase map $\varphi_4^{(1)}$ of the 4th phase plate is not accurate. However, as iteration number i increases, $E(\boldsymbol{\rho}, z_0; \boldsymbol{\rho}_1, z_4)$, and $E(\boldsymbol{\rho}, z_0; \boldsymbol{\rho}_2, z_4)$ have revealed a clear target structure in the amplitude part while their phase maps show spherical wavefronts associated with the corresponding positions, $\boldsymbol{\rho}_1$ and $\boldsymbol{\rho}_2$. Also, the quantified phase map $\varphi_4^{(i)}$ gradually converged to the ground-truth phase map shown in Fig. 3b.

Figure R2. Detailed iteration process of finding phase plates. **a**, Illustration of $E(\boldsymbol{\rho}, z_0; \boldsymbol{\rho}_j, z_4)$ in Eq. (5), for the 4th layer of numerical simulation condition in Fig. 3. **b**, Phase map of the Green's function $G(\boldsymbol{\rho}, z_0; \mathbf{0}, z_4)$, describing the spherical wavefront of point source propagating a distance of $z_4 - z_0$. **c**, Each panel shows two representative images of the columns of \mathbf{R}_{z_0, z_4} in their amplitude (upper) and phase (middle) for iteration number $i = 1, 2$, and 10, respectively, from the numerical simulation results in Fig. 3. We also present the quantified phase map at each iteration step. $\varphi_4^{(i)}$ is the phase correction i^{th} iteration step, and $\varphi_4^{(c)}$ is the accumulated phase correction up to the i^{th} iteration step.

To clarify that the effect of M diminishes with iteration, we added the following sentences below Eq. (5).

“Here, M represents the contribution of multiple scattering caused by the other layers. As we will discuss in the following and demonstrate in Supplementary Section 3, the influence of this factor gradually diminishes as the number of iterations increases.”

614: Eq. (6) and Eq. (S15) are actually valid on the support of object \mathbf{O} . If the object is scarce as for the myelinated axons, how does it affect the convergence of the phase correction?

As pointed out by the reviewer, the convergence of MST algorithm relies on the support of the object \mathbf{O} . A short answer to the reviewer's question is that our algorithm demonstrates robust performance, even with target objects that present extremely weak contrast. In this context, the object of interest includes not only the amplitude reflectance from the myelinated axons but also from the inhomogeneous tissues within the depth section set by the objective focus and time gating. In other words, the weak specular reflections from tissue inhomogeneities serve as the object function as well. Therefore, the proposed method remains effective in areas where myelinated axons are scarce.

Here is a detailed explanation. The MST algorithm fails if the initial correlation supported by \mathbf{O} is too weak or the contribution of multiple scattering \mathbf{M} by other layers is too significant. However, we gain

the initial correlation from a large number of measurements in the reflection matrix \mathbf{R} . Although we described the phase quantification process in Eq. (S15) as if we compare only two column vectors of propagated reflection matrix \mathbf{R}_{z_0, z_k} , the actual algorithm used in our study utilizes the whole column vectors to determine the phase function with substantially enhanced the fidelity of convergence.

More specifically, in the actual algorithm, we define a Green's function normalized matrix \mathbf{S}_{z_0, z_k} whose elements are given by

$$\mathbf{S}_{z_0, z_k}(\boldsymbol{\rho}, z_0; \boldsymbol{\rho}', z_k) \equiv \frac{E(\boldsymbol{\rho}, z_0; \boldsymbol{\rho}', z_k)}{G(\boldsymbol{\rho} - \boldsymbol{\rho}'; z_0 - z_k)} = e^{i\varphi_k(\boldsymbol{\rho}')} \mathbf{O}(\boldsymbol{\rho}, z_0) + \mathbf{M}'(\boldsymbol{\rho}; \boldsymbol{\rho}'). \quad (\text{R1})$$

Then we quantify the phase function of k^{th} phase plate by solving

$$\arg \min_{\boldsymbol{\varphi}_k} \|\mathbf{S}_{z_0, z_k} - \tau \boldsymbol{o}^T \times e^{i\boldsymbol{\varphi}_k}\|_2, \quad (\text{R2})$$

where \boldsymbol{o}^T is a transpose of the vectorized object function (column vector), $e^{i\boldsymbol{\varphi}_k}$ is vectorized phase function of the corresponding phase plate (row vector), and $\|\mathbf{A}\|_2$ implies the Frobenius norm of a matrix \mathbf{A} . This minimization process is identical to finding the first singular vector of \mathbf{S}_{z_0, z_k} , and we utilized the well-known power iteration method (Nash, J.C., et. al., *The Computer Journal*, **30** 268, 1987).

Consequently, the support of the object \mathbf{O} in the initial step of phase quantification can be maximized since we utilize the whole measured matrix in the process of Eq. (R2). For example, the through-skull imaging in Fig. 6 measures the reflection matrix through 160×160 sampling points across the x- and y-axis, and the number of column vectors in Eq. (R2) becomes 25,600. We extract the common object information as well as the phase function of the corresponding layer from this huge number of column vectors with Eq. (R2). As a result, despite the weak reflectivity of myelinated axons and the severe multiple scattering by 200 μm thick skull tissue, we could successfully apply the MST algorithm as presented in Fig. 6.

We added the following sentences to the Discussion section to emphasize that the proposed method works for real biological samples having extremely weak reflectance contrast.

“Our proposed method works even for objects with extremely weak contrast such as myelinated axons and inhomogeneous tissue textures inside the brain. This is because the algorithm finds correlations among common object function contained in a large number of complex fields constituting the reflection matrix.”

We also added the discussions made in the response, including Eqs. (R1) and (R2), to Methods section and Supplementary Section 2.2.

673-674: typo? 'The saline covering the skull' I guess the authors meant the *scalp* covering the skull.

Revise accordingly.

677: typo 'to keep keeping'

Revise accordingly.

supp inf:section 2.2: MW is not defined after Eq.(S12).

Revise accordingly.

Reviewer #2

Report on “Tracing multiple scattering trajectories for deep optical imaging in scattering media” This article deals with imaging through disordered media that scatter light multiple times. Such distortions due to wave scattering constitute a major bottleneck for bio-medical imaging as traditional approaches typically fail in this context. Here, an approach is proposed and implemented that mimics a scattering medium through multiple complex phase plates, whose phase distortions are found computationally (in post-processing of reflection matrix measurements).

I find this work conceptually appealing and of interest to the community working on multiple scattering, inverse problems, and imaging techniques. There are, however, also several weak points that need to be addressed by the authors:

We are grateful to the reviewer for highly evaluating the conceptual novelty of our work and recognizing its significant contribution to the field of deep imaging. We also thank the reviewer for the thoughtful questions and suggestions that enable us to enhance the clarity in conveying the concepts and methodologies. In this response letter, we have addressed all the issues raised by the reviewer in detail. We have also explained why our proposed method works for targets with extremely weak contrast and provided additional analysis to support this explanation. An essential strength of our MST algorithm is that it enables the solving of high-order inverse scattering problems for *in vivo* through-skull deep brain imaging, thereby ensuring a very high level of practicality.

1. The idea to compensate multiple forward-scattering through a cascade of phase masks is not new (see arXiv:2204.02865). Also the compensation of multiple scattering (both in forward and backward direction) through a physically fabricated anti-reflection structure has recently been presented (see Nature 607, 281 (2022)). With the first work being numerical only and the second one being implemented in the microwave domain, I still see room for the present work in the literature, but it clearly needs to be better embedded in the state of the art. My impression is that also recent work by Aubry et al. (arXiv:2303.06119 & arXiv:2303.07483) has a certain closeness to the current manuscript that deserves to be discussed.

We agree with the reviewer that the concept of modeling a scattering medium as a cascade of phase masks is not novel. Indeed, constructing such a model is simply a matter of establishing a set of equations to solve. The true challenge lies in identifying multiple specific phase masks that accurately mimic the scattering properties of the original medium. Of even greater importance is determining whether this methodology allows us to detect an object within the scattering medium. It's in this context that the innovative aspect of our work becomes clear. Our method does more than just identify multiple phase masks from an experimentally recorded reflection matrix in an epi-detection geometry, a critical aspect for *in vivo* imaging. Importantly, the identified phase masks embody the actual physical structures of the scattering medium. This subsequently enables the computational elimination of these structures, thereby revealing the embedded object.

The papers that the reviewer mentioned are somewhat related to our study, but fundamentally, their goal is not imaging. The study presented in arXiv:2204.02865 aims to find multiple phase masks that produce a transmission matrix similar to that of a multimode optical fiber. Another paper (Nature 607, 281 (2022)) reports a method for finding structures that generate a reflection matrix satisfying a critical coupling condition. In both studies, the discovered phase masks or structures do not relate to the internal structure of the actual scattering medium. Thus, they do not reveal an object within the scattering medium. However, they share a commonality with our work in finding structures that possess similar scattering characteristics to the measured transmission/reflection matrix. We acknowledge that the methods discussed in these reference papers could be valuable for future research. As such, we added the following sentence to the Discussion section.

“Other approaches to finding the multiple phase masks or designed structures that produce a similar scattering matrix as the measured one could provide another possibility, although their scope is different from deep imaging (Nature 607, 281 (2022), arXiv:2204.02865).”

The recent studies from the Aubry group (arXiv:2303.06119 & arXiv:2303.07483) are indeed intriguing and merit attention. Essentially, these studies use an iterative time reversal algorithm on the distortion matrix transformed from the reflection matrix, seeking to identify the sample-induced aberrations. Their scope is similar to our previously developed CLASS algorithm, which decomposes the reflection matrix into output aberration, object function, and input aberration. The scope of the present study is beyond all these earlier studies as we correct multiple scattering *in situ* by identifying multiple phase plates that describe the scattering medium from the measured reflection matrix. As the recent studies from the Aubry group fall under the category of third-order inverse scattering problems, we have referenced them in the introduction alongside other studies on aberration correction.

2. What I clearly missed in the main text of the manuscript is an easily accessible explanation of how the MST algorithm works and how it is able to determine the required degrees of freedom in the phase masks. (I also found the explanation in the methods section hard to follow.) Maybe it would also be helpful in this context to include an explanatory figure that illustrates the basic principles of the algorithm. Also the new method's limitations should be more clearly spelled out in terms of real-world applicability.

In our original manuscript, we aimed to primarily deliver key concepts in the main text while reserving a detailed explanation of the MST algorithm for the Methods section and Supplementary Section 2.2. Following the reviewer's suggestion, we have now included an explanatory figure and its description in the Methods section to better elucidate the working principle of the MST algorithm.

“Next, we elaborate on the key concept to identify $\varphi_k(\boldsymbol{\rho}_j)$ from the measured reflection matrix. The first term on the right-hand side of Eq. (3) contains the object's amplitude reflectance function $O(\boldsymbol{\rho}_i, z_0)$ and the free-space Green's function. In Figs. 7a-b, we illustrate the first term in Eq. (3) of the two electric fields $E_1(\boldsymbol{\rho}_i, z_0; \boldsymbol{\rho}_1, z_k)$ and $E_2(\boldsymbol{\rho}_i, z_0; \boldsymbol{\rho}_2, z_k)$ for two different source positions $P_1(\boldsymbol{\rho}_1, z_k)$ and $P_2(\boldsymbol{\rho}_2, z_k)$, respectively. The two fields resemble each other, except for the center of the spherical phase profile and the overall phase retardation. After normalizing out the Green function in each field, we compute the field-field correlation to determine the relative phase retardations $\varphi_k(\boldsymbol{\rho}_2) - \varphi_k(\boldsymbol{\rho}_1)$ (Fig. 7b). The presence of E_M with its magnitude much larger than the first term makes it difficult to find $\varphi_k(\boldsymbol{\rho}_j)$ accurately and gives rise to an error. As we shall explain below, the field correlation across the object plane raises the fidelity of finding $\varphi_k(\boldsymbol{\rho}_j)$, because the first term of Eq. (3) is coherently added with respect to the reference point at $\boldsymbol{\rho}_{\text{ref}}$, whereas E_M is incoherent in the estimation of the correlation. Thus, the fidelity is determined by the number of detection pixels in the object plane.”

Figure 7. Working principle of the quantification of $\varphi_k(\boldsymbol{\rho})$ in MST algorithm. **a**, Illustration of the first term of $E(\boldsymbol{\rho}_i, z_0; \boldsymbol{\rho}_j, z_k)$ in Eq. (3), which is the j^{th} column vector of the propagated reflection matrix \mathbf{R}_{z_0, z_k} with $j = 1, 2$. **b**, Illustration of phase correlation method for finding the phase functions $\varphi_k(\boldsymbol{\rho}_j)$. Examples of $E(\boldsymbol{\rho}_i, z_0; \boldsymbol{\rho}_1, z_k)$ and $E(\boldsymbol{\rho}_i, z_0; \boldsymbol{\rho}_2, z_k)$ are shown at the first row. After normalizing the respective Green's functions (second row), only the object function remains along with $\varphi_k(\boldsymbol{\rho}_1)$ and $\varphi_k(\boldsymbol{\rho}_2)$. Based on the correlation of the two field images in the third row, an approximate phase function $\varphi_k(\boldsymbol{\rho}_2) - \varphi_k(\boldsymbol{\rho}_1)$ can be obtained. All the images in **d** represent phase maps. Note that the illustrations in **a** and **b** show only the first term in Eq. (3) for clarity.

We have demonstrated the potential applicability of our MST algorithm to real-world scenarios by proving its efficacy for *in vivo* through-skull imaging (Fig. 6). Nevertheless, we acknowledge that there is ample room for further refinement in future studies. In fact, we discuss the limitations of our newly proposed method and its outlook in the Discussion section of the original manuscript:

“Our algorithm finds multiple scattering trajectories based on the wave correlation of multiple scattering having interacted with a target object. This approach is intuitive, robust, and cost-effective. However, this doesn’t trace all the multiple scattering containing the object information. One may consider combining the MST algorithm with other computational approaches such as compressive sensing and deep learning to increase the traceable multiple scattering trajectories. Imaging geometry is another defining factor, and one can consider various collection geometries to better capture the multiple scattering of interest. The width of time gating is shortened as much as possible in conventional imaging to better rule out the multiple scattering, but there may be an optimal time gating window for collecting useful multiple scattering. The extension of the algorithm to incorporate backscattering in the scattering medium can be another important direction. Future studies addressing all these factors will extend the degree of multiple scattering coverage.”

3. I stumbled over the following sentence: “In the first strategy, we measure the electric fields of the backscattered waves, including both the ballistic wave and multiply scattered waves, whose flight times are identical to that of the ballistic wave.” How can it be that a wave that is scattered can have the same flight time as a ballistic wave, especially when it also needs to reach the target? Doesn’t scattering necessarily lead to an increase in the flight time as compared to the straight-line path of a ballistic wave?

As the reviewer pointed, multiply scattered waves exhibit longer optical pathlength, leading to increased flight times compared to the straight-line path of the ballistic waves. In our experiment, we applied a finite time-gating window, which is determined by the pulse width of the laser (100 fs, equivalent to 25 μm in length). This time-gating window allows us to selectively detect the reflection signal with flight times falling within this defined interval. To avoid the ambiguity pointed out by the reviewer, we revised the main text as follows.

“In the first strategy, we measure the electric fields of the backscattered waves, including both the ballistic wave and multiply scattered waves, whose flight times are distributed within a finite time-gating window of 100 fs, given by the pulse width of the laser.”

4. Which restrictions do apply to the target? The authors use a specific target here with a high contrast between bright and dark parts. Would their approach also be applicable to objects with a weak contrast?

While we initially used a resolution target for proof-of-concept experiments (Figs. 4 and 5), we have also demonstrated the applicability of the MST algorithm to *in vivo* through-skull deep brain imaging (Fig. 6). Our algorithm exhibits robust performance, even when applied to target objects with extremely weak contrast. In deep brain imaging, the object of interest comprises not only the amplitude reflectance from the myelinated axons but also from the inhomogeneous tissues within the depth section set by the objective focus and time gating. In other words, the weak specular reflections from tissue inhomogeneities also function as an object. As a result, the proposed method retains its effectiveness in any area within living tissues.

To be more specific, the convergence condition of the MST algorithm can be discussed in two aspects.

1. The convergence condition depends on the initial correlation between the columns of the propagated reflection matrix \mathbf{R}_{z_0, z_k} after normalizing the corresponding spherical wavefront, as discussed in Eq. (S14).

2. Due to the imaging geometry, there is a finite spatial resolution in the reconstruction of phase plates, as discussed in Supplementary Section 2.2. As a result, the MST algorithm cannot trace multiple scattering by scatterers smaller than the reconstruction resolution. In other words, the convergence condition also depends on the type of scattering medium.

The reflectance of the target objects can influence the convergence of the MST algorithm by the first aspect. If the reflectance of the target object is too weak, the contribution of multiple scattering may dominate the initial correlation provided by the object, leading to the failure of convergence. However, it's important to clarify that the MST algorithm does not necessarily require high contrast. The key factor for the MST algorithm's success is in utilizing the phase correlation provided by the target object among a large number of complex fields constituting the reflection matrix. As long as the target objects remain unchanged during the matrix measurement, the contrast of the target object does not significantly affect the performance of the MST algorithm.

To illustrate this point, we applied the MST algorithm to a target object lacking distinct bright and dark signal contrast, even without any myelinated axons, as shown in Fig. R4. Specifically, we used the MST algorithm on the upper part of a mouse brain (210 μm) from the same mouse used in Fig. 6. Despite the absence of high-contrast structures such as myelinated axons at the depth, the performance of identifying the phase functions for the skull tissue was comparable to the results obtained in Fig. 6. This demonstration supports our claim that the MST algorithm can still perform effectively in scenarios where high-contrast structures are not present.

Figure R4. Demonstration of MST algorithm with a low contrast intra-brain structures through an intact skull. **a-b**, Confocal reflectance image and MST image of the mouse brain at a depth 210 μm from the surface of the skull, respectively. We imaged the upper part of the mouse brain from a live mouse which was used in the experiment of Fig. 6. Scale bar, 30 μm . Color scales are normalized by the peak intensity of **b**. **c**, Identified phase functions at five distinct layers located by {80, 120, 160, 200, 240} μm above the target plane, respectively. Scale bar, 50 μm . The lengths of scale bars differ because the size of each phase plate's reconstructed area was determined differently by its distance from the target plane.

We added the following sentences to the Discussion section to emphasize that the proposed method works for real biological samples having extremely weak reflectance contrast.

“Our proposed method works even for objects with extremely weak contrast such as myelinated axons and inhomogeneous tissue textures inside the brain. This is because the algorithm finds correlations among common object function contained in a large number of complex fields constituting the reflection matrix.”

5. It isn't clear to me how the authors determine the number of scattering events that they can compensate for (like 17 in the conclusion section).

In the MST model, we approximate the scattering medium with a series of phase plates. Determining the appropriate number of phase plates for this approximation is not simple, as it depends on the imaging geometry and various structural properties of the scattering medium. However, the maximum number of phase plates that our algorithm can identify is largely determined by the achievable axial resolution Δz_k of quantifying φ_k . As explained in the Methods section, it is given by $\Delta z_k = (2 z_k / L_0)^2$, where L_0 is the side length of the field of view. The number of phase plates can be increased up to the point that their spacings correspond to the achievable axial resolution set by the imaging geometry. Other factors may include the thickness of the scattering medium, average scatterer size, and target object type. However, it's clear that a larger number of phase plates will yield more accurate results, although this demands greater computational power, as illustrated in Fig. 5f. In our experiment, we sought to find the optimum number of layers by increasing the layer number N from 1 to 8, as shown in Fig. 5. As a result, we observed that performance improvement began to saturate after $N=5$, despite a steady increase.

To elaborate on the number of 'scattering events' we compensate for in the MST model, each phase plate induces either a single scattering event or none at all. For instance, if a phase plate is similar to a thin diffraction grating, the first-order diffraction corresponds to a single scattering event, and the zeroth-order diffraction corresponds to no interaction. Consequently, when modeling a scattering medium with 8 discrete layers, up to 17 scattering events can occur, accounting for wave propagation on the way in and out, and the reflection by the target object.

We added the following sentences to the Discussion section to clarify how the number of layers that MST algorithm can trace is determined.

“Imaging geometry is another defining factor. It determines the lateral and axial resolutions of the layers that the algorithm can identify (see details in the Methods section). Consequently, it largely defines the maximum number of layers or the maximum number of scattering events that our algorithm can trace.”

6. It may also require further explanation as to why the method can ultimately be applied to systems with very pronounced wave backscattering (such as the mouse skull).

As the reviewer commented, our method works with very pronounced wave backscattering, which is not accounted for in our forward model. This is mainly because of the time gating, which rules out a large fraction of the pronounced wave backscattering occurring in the upper part of the tissue having no interaction with the target object at the depth of interest. To clarify this, we added a sentence explaining the role of time gating in the ‘Working principle’ section.

“In the first strategy, we measure the electric fields of the backscattered waves, including both the ballistic wave and multiply scattered waves, whose flight times are distributed within a finite time-gating window of 100 fs, given by the pulse width of the laser. This eliminates a large fraction of scattering events that have no interaction with the target objects located at the depth of interest. These include backscattering at shallow depths.”

As mentioned in the Discussion section, our algorithm currently does not trace backscattering occurring at the phase plates modeling the scattering medium. In the next stage of research, we plan to broaden the width of the time-gating window and aim to trace multiply backscattered waves at the scattering medium that interact with the target object. We have outlined this future direction in the Discussion section.

7. In Eq. 6 the “arg” seems to appear twice.

We revised the equation accordingly.

In summary, this is a very interesting contribution to the emerging field of matrix imaging. After the authors have had a chance to address the points raised above, I would be glad to have another look at the manuscript.

Once again, we appreciate the reviewer's thoughtful questions and suggestions. By addressing them in full, we believe that the clarity of the paper has been greatly enhanced.

Reviewer #1 (Remarks to the Author):

The authors addressed all concerns from both reviewers, and significantly improved the paper. The MST algorithm is now better introduced, and key features and assumptions are more clearly identified and explained.

I therefore recommend for publication.

Reviewer #2 (Remarks to the Author):

The authors have answered all the questions I asked in my report and implemented corresponding modifications to their manuscript. I am satisfied with the new paper version and recommend it for publication in Nature Communications.